# Developmental conversion of thymocyte-attracting cells into self-antigen-displaying cells in embryonic thymus medulla epithelium

**Izumi Ohigashi[1], Andrea J White[2], Mei-Ting Yang[3], Sayumi Fujimori[1], Yu Tanaka[3†], Alison Jacques[3], Hiroshi Kiyonari[4], Yosuke Matsushita[5], Sevilay Turan[6], Michael C Kelly[7], Graham Anderson[2], Yousuke Takahama[3]***

[1]Division of Experimental Immunology, Institute of Advanced Medical Sciences, University of Tokushima, Tokushima, Japan; [2]Institute for Immunology and Immunotherapy, University of Birmingham, Birmingham, United Kingdom; [3]Thymus Biology Section, Experimental Immunology Branch, National Cancer Institute, National Institutes of Health, Bethesda, United States; [4]Laboratory for Animal Resources and Genetic Engineering, RIKEN Center for Biosystems Dynamics Research, Hyogo, Japan; [5]Division of Genome Medicine, Institute of Advanced Medical Sciences, University of Tokushima, Tokushima, Japan; [6]Sequencing Facility, Frederick National Laboratory for Cancer Research, National Cancer Institute, Frederick, United States; [7]Single Cell Analysis Facility, Cancer Research Technology Program, National Cancer Institute, National Institutes of Health, Bethesda, United States

**\*For correspondence:**
yousuke.takahama@nih.gov

**Present address:** †Department of Pediatrics, The University of Tokyo Hospital, Hongo, Japan

**Competing interest:** The authors declare that no competing interests exist.

**Abstract** Thymus medulla epithelium establishes immune self-tolerance and comprises diverse cellular subsets. Functionally relevant medullary thymic epithelial cells (mTECs) include a self-antigen-displaying subset that exhibits genome-wide promiscuous gene expression promoted by the nuclear protein Aire and that resembles a mosaic of extrathymic cells including mucosal tuft cells. An additional mTEC subset produces the chemokine CCL21, thereby attracting positively selected thymocytes from the cortex to the medulla. Both self-antigen-displaying and thymocyte-attracting mTEC subsets are essential for self-tolerance. Here, we identify a developmental pathway by which mTECs gain their diversity in functionally distinct subsets. We show that CCL21-expressing mTECs arise early during thymus ontogeny in mice. Fate-mapping analysis reveals that self-antigen-displaying mTECs, including Aire-expressing mTECs and thymic tuft cells, are derived from CCL21-expressing cells. The differentiation capability of CCL21-expressing embryonic mTECs is verified in reaggregate thymus experiments. These results indicate that CCL21-expressing embryonic mTECs carry a developmental potential to give rise to self-antigen-displaying mTECs, revealing that the sequential conversion of thymocyte-attracting subset into self-antigen-displaying subset serves to assemble functional diversity in the thymus medulla epithelium.

## eLife assessment

This **important** study provides new insights into the development and function of medullary thymus epithelial cells (mTEC). The authors provide **compelling** evidence to support their claims as to the differentiation and lineage outcomes of CCL21+ mTEC progenitors, which further our understanding of how central tolerance of T cells is enforced within the thymus.

## Introduction

The thymus medulla epithelium provides a unique environment for newly generated T cells to acquire self-tolerance before their emigration to the circulation. A variety of medullary thymic epithelial cells (mTECs) contribute to displaying self-antigens by employing multiple mechanisms. Promiscuous gene expression by a subpopulation of mTECs that express the nuclear protein Aire allows chromatin remodeling and genome-wide ectopic gene transcription (*Derbinski et al., 2001*; *Anderson et al., 2002*). Cellular mimesis in other mTEC subpopulations relies on a mosaic of tissue-specific transcription factor-dependent cells that individually resemble specialized cell types, including mucosal tuft cells and microfold cells (*Farr et al., 2002*; *Michelson et al., 2022*). These mTEC subpopulations form a self-antigen-displaying mTEC subset that provides various self-components, including extrathymic tissue-specific molecules, in the thymus medulla, so that newly generated T cells have an opportunity to establish central tolerance to genome-wide and systemic self-antigens prior to export from the thymus.

In addition to the self-antigen-displaying mTEC subset, there exists a medullary epithelial subset that produces the C–C chemokine CCL21, which attracts developing thymocytes and dendritic cells to the medullary region (*Takahama et al., 2017*). Positive selection-inducing TCR signals in newly generated cortical thymocytes elevate the expression of CCR7, a receptor for CCL21, and positively selected cortical thymocytes are therefore specifically attracted to migrate to the medullary region through the CCL21–CCR7-mediated chemotactic signals (*Ueno et al., 2004*; *Takahama, 2006*). Two molecular species CCL21Ser and CCL21Leu with one amino acid difference are encoded in the mouse genome (*Nakano and Gunn, 2001*; *Chen et al., 2002*), and CCL21Ser encoded by *Ccl21a* locus is predominantly expressed in the thymus medulla (*Kozai et al., 2017*). In mice lacking either *Ccr7* or *Ccl21a*, positively selected mature thymocytes fail to accumulate in the medulla, and therefore T cells fail to establish self-tolerance (*Kozai et al., 2017*; *Kurobe et al., 2006*; *Nitta et al., 2009*). Additional CCR7-ligand CCL19 has no appreciable role in the cortex-to-medulla migration of developing thymocytes (*Kozai et al., 2017*; *Link et al., 2007*). Thus, CCL21-expressing thymocyte-attracting mTECs represent a functional mTEC subset that is essential for the establishment of self-tolerance in T cells.

Aire-expressing mTECs primarily belong to MHC-II$^{high}$ CD80$^{high}$ mTEC subpopulation (mTEC$^{high}$) (*Derbinski et al., 2005*; *Gray et al., 2006*), whereas most thymic mimetic cells are detectable in MHC-II$^{low}$ CD80$^{low}$ mTEC subpopulation (mTEC$^{low}$), which are derived from Aire-expressing mTEC$^{high}$ cells (*Michelson et al., 2022*; *Miller et al., 2018*; *Kadouri et al., 2020*; *Metzger et al., 2013*). The mTEC$^{low}$ subpopulation is heterogeneous and contains immature mTEC progenitors that differentiate into the Aire-expressing mTEC$^{high}$ subpopulation (*Rossi et al., 2007*; *Gäbler et al., 2007*). Consequently, the developmental progression of the self-antigen-displaying mTEC subset is considered to occur in the following sequence: mTEC$^{low}$ progenitors -> mTEC$^{high}$ Aire-expressing cells -> mTEC$^{low}$ mimetic cells.

CCL21-expressing mTECs are mostly included in the mTEC$^{low}$ subpopulation (*Lkhagvasuren et al., 2013*). However, how the CCL21-expressing mTEC subset is integrated with, and/or potentially diverted from, the developmental progression of the self-antigen-displaying mTEC subset remains unknown. A single-cell RNA-sequencing-based cluster analysis reported that *Ccl21a* transcripts are detectable within a heterogeneous 'intertypical' TEC subpopulation, which includes cells with gene expression profiles resembling immature mTEC progenitors (*Baran-Gale et al., 2020*). A more recent study has indicated that postnatally appearing mTEC-biased progenitors include cells that transcribe *Ccl21a* (*Nusser et al., 2022*). However, an independent single-cell RNA-sequencing-based study suggested that TECs expressing *Ccl21a* transcripts represent a terminal population derived from 'transit-amplifying' mTEC progenitors, by branching out of the self-antigen-displaying mTEC lineage (*Wells et al., 2020*). Without the direct evaluation of developmental potential, it remains controversial and unknown how CCL21-expressing mTECs are developmentally related to immature mTEC progenitors. It is even unclear how the *Ccl21a*-transcript-detectable TECs reported in these single-cell transcriptomic studies are related to functionally relevant CCL21-protein-producing medullary TECs. More specifically, it remains unestablished whether the functionally relevant CCL21-expressing mTEC subset retains a developmental potential to give rise to the self-antigen-displaying mTEC subset.

The present study addresses whether the CCL21-expressing mTEC subset carries a developmental capability to differentiate into the self-antigen-displaying mTEC subset. We show that CCL21-expressing mTECs arise early during embryonic development in mice. By Cre-loxP-mediated fate-mapping analysis, we demonstrate that a vast majority of self-antigen-displaying mTEC subset,

including Aire-expressing mTECs and thymic tuft cells, originates from CCL21-expressing cells. The developmental potential of CCL21-expressing embryonic mTECs to give rise to Aire-expressing mTECs is confirmed in reaggregate thymus experiments. These results indicate that functionally relevant CCL21-expressing mTEC subset contains a developmental capability to differentiate into self-antigen-displaying mTEC subset and that developmental conversion from thymocyte-attracting subset to self-antigen-displaying subset serves to assemble functional diversity in thymus medulla epithelium.

## Results

### CCL21-expressing mTECs arise in embryonic thymus

We began this study by examining the appearance of CCL21-expressing mTECs during mouse ontogeny. To do so, we analyzed CCL21-expressing mTECs by detecting tandem dimer Tomato (tdTomato) fluorescence proteins in *Ccl21a-tdTomato* knock-in mice. In the postnatal thymus of 5-week-old *Ccl21a*$^{tdTomato/+}$ mice, CCL21-expressing cells were detected in keratin 14 (Krt14)-expressing medullary TECs (*Figure 1A*) and not in Ly51-expressing cortical TECs (*Figure 1B*). Flow cytometric analysis confirmed that Ccl21a$^{tdTomato+}$ cells in the thymus are predominantly confined in EpCAM$^+$Ly51$^-$UEA1$^+$ mTECs (*Figure 1C, D*). In accordance with the medullary localization of CCL21-expressing mTECs, thymic medulla was abundant in CCL21 proteins (*Figure 1E*), which contribute to attracting CCR7-expressing positively selected thymocytes from the cortex (*Takahama, 2006*; *Kozai et al., 2017*). Among mTECs, most CCL21-expressing mTECs were distinct from Aire-expressing mTECs and DCLK1$^+$ thymic tuft cells (*Figure 1F*). CCL21-expressing mTECs equally distributed throughout the medullary region and were enriched neither in the pericortical area, including the cortico-medullary junction area, nor in the central area within the medulla (*Figure 1G, H*, *Figure 1—figure supplement 1*).

Analysis of embryonic thymus showed that CCL21-expressing cells in the central region of the thymus were prominent by embryonic day 15 (E15) (*Figure 2A*) and detectable as early as E13 (*Figure 2B*). The localization of CCL21-expressing cells in the central region of embryonic thymus lobes and their expression of Krt14 and Foxn1 ascertained that CCL21-expressing embryonic thymus cells represented mTECs (*Figure 2A–C*), rather than neighboring CCL21$^+$ parathyroid gland cells (*Figure 2B*), which contribute to the attraction of hematopoietic cells to the pre-vascularized embryonic thymus (*Liu et al., 2006*). CCL21 proteins were detectable in the medullary region proximal to Ccl21a$^{tdTomato}$-expressing mTECs in the embryonic thymus by E15 (*Figure 2D*), indicating that Ccl21a$^{tdTomato}$-expressing E15 embryonic mTECs represent functionally relevant CCL21-expressing mTECs. On the other hand, CCL21-expressing mTECs in the embryonic thymus were largely distinct from Aire-expressing mTECs and DCLK1$^+$ thymic tuft cells (*Figure 2E*). Similar to the postnatal thymus, CCL21-expressing mTECs in the embryonic thymus distributed throughout the medullary region and were not enriched in the pericortical areas including the cortico-medullary junctions (*Figure 2F, G*, *Figure 2—figure supplement 1*). These results indicate that CCL21-expressing mTECs arise by E15 in the central region of the embryonic thymus.

### Fate mapping of CCL21-expressing cells in thymus

To better understand whether CCL21-expressing mTECs have a developmental potential to give rise to the self-antigen-displaying mTEC subset, we engineered mice in which the P1 bacteriophage-derived *Cre* recombinase gene was knocked-in at *Ccl21a* locus (*Figure 3A*). We bred *Ccl21a*$^{Cre/+}$ mice with mice that carried a ubiquitous CAG promoter-driven loxP-stop-loxP-EGFP (CAG-loxP-EGFP) transgene. In *Ccl21a*-Cre × CAG-loxP-EGFP mice, EGFP expression labeled cells that have transcribed *Ccl21a* previously and/or presently (*Figure 3A*).

In the postnatal thymus of 3-week-old *Ccl21a*-Cre × CAG-loxP-EGFP mice, EGFP signals in the thymus distributed prominently in the medullary region (*Figure 3B*). EGFP+ cells in the medulla were detected in Krt14$^+$ mTECs (*Figure 3B*), including Aire$^+$ mTECs and DCLK1$^+$ thymic tuft cells (*Figure 3C*). Flow cytometric analysis of enzyme-digested thymus cells showed that EGFP$^+$ cells in the thymus were detected exclusively in EpCAM$^+$ CD45$^-$ TECs and not in EpCAM$^-$CD45$^+$ hematopoietic cells, including thymocytes and dendritic cells, or EpCAM$^-$CD45$^-$ cells, including mesenchymal fibroblasts and endothelial cells (*Figure 3D and E*). Within EpCAM$^+$CD45$^-$ TECs, approximately 95% of

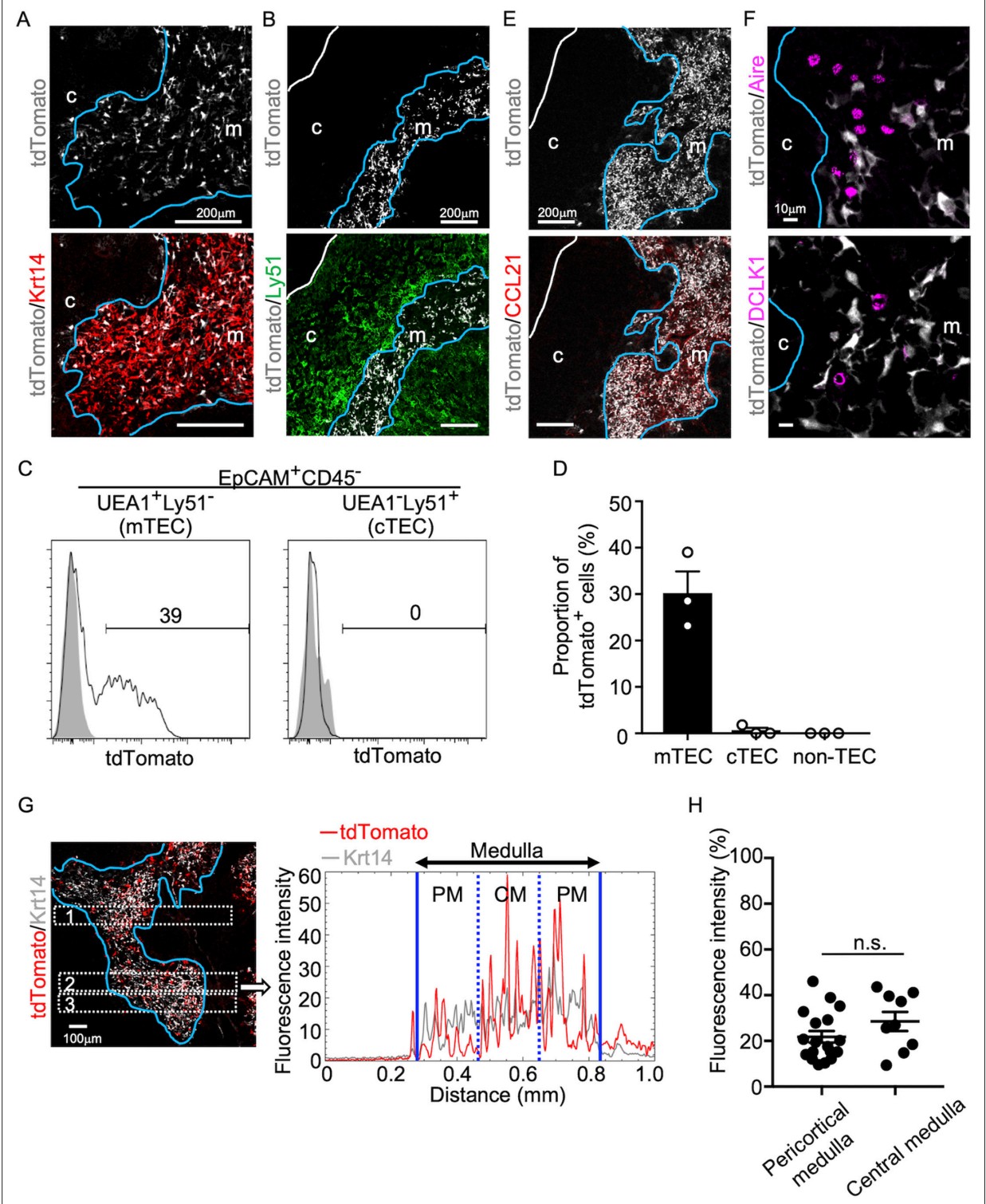

**Figure 1.** CCL21-expressing medullary thymic epithelial cells (mTECs) in postnatal thymus. Immunofluorescence analysis of thymus sections from 5-week-old *Ccl21a^{tdTomato/+}* mice. tdTomato fluorescence (white) was detected with Keratin 14 (Krt 14, red) (**A**) or Ly51 (green) (**B**). White lines indicate capsular outline of the thymus. Blue lines show cortico-medullary junction. Representative data from at least three independent experiments are shown. c, cortex. m, medulla. Scale bar, 200 µm. (**C**) Flow cytometric profiles of tdTomato fluorescence expression in EpCAM^+CD45^−UEA1^+Ly51^− mTECs (left) and EpCAM^+CD45^−UEA1^−Ly51^+ cTECs (cortical thymic epithelial cells) (right) from *Ccl21a^{tdTomato/+}* mice (black line histograms) and WT mice (gray shaded histograms). Numbers in histograms indicate frequency of cells within indicated area. (**D**) Proportions (means and standard error of the means [SEMs], *n* = 3) of tdTomato^+ cells in EpCAM^+CD45^−UEA1^+Ly51^− mTECs, EpCAM^+CD45^−UEA1^−Ly51^+ cTECs, and EpCAM^− non-TECs analyzed as in (**C**). (**E,**

*Figure 1 continued on next page*

*Figure 1 continued*

**F**) Immunofluorescence analysis of thymus sections from 5-week-old *Ccl21a*$^{tdTomato/+}$ mice. tdTomato fluorescence (white) was detected with CCL21 (red) (**E**), Aire, or DCLK1 (magenta) (**F**). White lines indicate capsular outline of the thymus. Blue lines show cortico-medullary junction. Representative data from at least three independent experiments are shown. c, cortex. m, medulla. Scale bar, 200 μm (**E**) or 10 μm (**F**). (**G**) Distribution of tdTomato-expressing cells in the medullary region defined by Krt14 expression. Left panel shows representative tdTomato (red) and Krt14 (white) fluorescence signals in the thymus sections from *Ccl21a*$^{tdTomato/+}$ mice. Scale bar, 100 μm. Right panel shows fluorescence intensity profiles of tdTomato (red line) and Krt14 (gray line) signals within the region of interest (ROI) defined by dashed rectangles in the left panel. Medullary regions were equally divided into three areas into pericortical areas (PM) and central area (CM). (**H**) tdTomato intensity (means and standard error of the means [SEMs], *n* = 3) in indicated areas (**G**) and *Figure 1—figure supplement 1A* were calculated in comparison with total tdTomato intensity within the ROI. n.s., not significant.

The online version of this article includes the following figure supplement(s) for figure 1:

**Figure supplement 1.** Distribution of *Ccl21a*-expressing cells in postnatal thymus.

UEA1$^+$Ly51$^-$ mTECs were EGFP$^+$, indicating that a vast majority of mTECs were derived from cells that previously transcribed *Ccl21a* (*Figure 3F, G*). Flow cytometric analysis also confirmed the detection of Aire$^+$UEA1$^+$Ly51$^-$ cells and DCLK1$^+$UEA1$^+$Ly51$^-$ cells in EpCAM$^+$CD45$^-$ EGFP$^+$ TECs (*Figure 3H*). Thus, self-antigen-displaying mTECs, including Aire$^+$ mTECs and DCLK1$^+$ thymic tuft cells, which do not presently express CCL21 (*Figures 1F, 2E*), originate from *Ccl21a*$^+$ progenitors.

In the quantitative RT-PCR analysis of *Cre*, *Ccl21a*, and other TEC-associated genes, such as *Aire*, *Tnfrsf11a*, and *Psmb11*, *Cre* was detected specifically in the *Ccl21a*-expressing mTEC$^{low}$ subpopulation, which includes the majority of CCL21-expressing mTECs (*Lkhagvasuren et al., 2013*), and not in the *Psmb11*-expressing cTECs or the *Aire-* and *Tnfrsf11a*-expressing mTEC$^{high}$ subpopulation (*Figure 4A*). These results verify the lineage tracing potential of *Ccl21a*-Cre-expressing cells to differentiate into the mTEC$^{high}$ subpopulation including Aire$^+$ mTECs.

In addition to mTECs, approximately 66% of UEA1$^-$Ly51$^+$ cTECs in the postnatal thymus were EGFP$^+$ in 3-week-old *Ccl21a*-Cre × CAG-loxP-EGFP mice (*Figure 3F, G*). Section analysis of the thymus verified the presence of EGFP$^+$ cells in the cortex (*Figure 3B*). Because cTECs do not presently express *Ccl21a* transcripts (*Kozai et al., 2017*) (also shown in *Figure 1B–D*), these results indicate that approximately two-thirds of postnatal cTECs are derived from cells that previously transcribed *Ccl21a* (*Figure 3F, G*). To characterize cTECs that previously transcribed *Ccl21a* and those that did not, we isolated EGFP$^+$ and EGFP$^-$ cTECs from 2-week-old postnatal *Ccl21a*-Cre × CAG-loxP-EGFP mice (*Figure 4—figure supplement 1*) and performed deep-sequencing transcriptomic analysis. The MA and volcano plots of RNA-sequencing data demonstrated high similarity in gene expression profiles between EGFP$^+$ and EGFP$^-$ cTECs among 20,230 genes detected (*Figure 4B, C*), unlike the pronounced difference in global gene expression profiles between cTECs and mTECs (*Figure 4—figure supplement 2A, B*). The genes that are relevant for T cell-inducing and positive selection-inducing functions in cTECs, including *Dll4*, *Il7*, *Cxcl12*, *Psmb11*, *Ctsl*, *Prss16*, *Cd83*, *H-2K*, *H-2D*, and *H2-Ab*, were similarly abundant in EGFP$^+$ and EGFP$^-$ cTECs (*Figure 4—figure supplement 2C*). These results suggest that cTECs derived from cells that either do or do not transcribe *Ccl21a* are functionally comparable to promote the generation and positive selection of cortical thymocytes.

Only 14 genes, including *Ccl21a*, showed significantly higher expression in EGFP$^+$ cTECs than in EGFP$^-$ cTECs, whereas only one gene, the parathyroid hormone-encoding *Pth*, showed significantly lower expression in EGFP$^+$ cTECs than in EGFP$^-$ cTECs (*Figure 4C*, *Figure 4—figure supplement 2D*). All the 14 genes that showed significantly higher expression in EGFP$^+$ cTECs than in EGFP$^-$ cTECs represented genes that showed higher expression in mTECs than in cTECs, and their expression, including *Ccl21a* expression, was much higher in mTECs than in EGFP$^+$ cTECs (*Figure 4—figure supplement 2D*). Interestingly, EGFP$^+$ cTECs distributed unequally within the thymic cortex and were detected more frequently in the perimedullary cortical region than in the central cortical region or the subcapsular region of the cortex (*Figure 4D, E*, *Figure 4—figure supplement 3A*). In contrast, Ly51$^+$ total cTECs did not show such an unequal distribution in the thymic cortex (*Figure 4—figure supplement 3B, C*). Thus, the intrathymic proximity to the medullary region contributes to the previous expression of *Ccl21a* transcripts in cTECs, suggesting that approximately two-thirds of postnatal cTECs originate from *Ccl21a*$^+$ cells generated in the thymic medulla.

Collectively, these fate-mapping experiments demonstrate that a vast majority of mTECs, including the self-antigen-displaying mTEC subset, and approximately two-thirds of cTECs, particularly cTECs in the perimedullary cortical region, are derived from progenitor cells that previously transcribed *Ccl21a*.

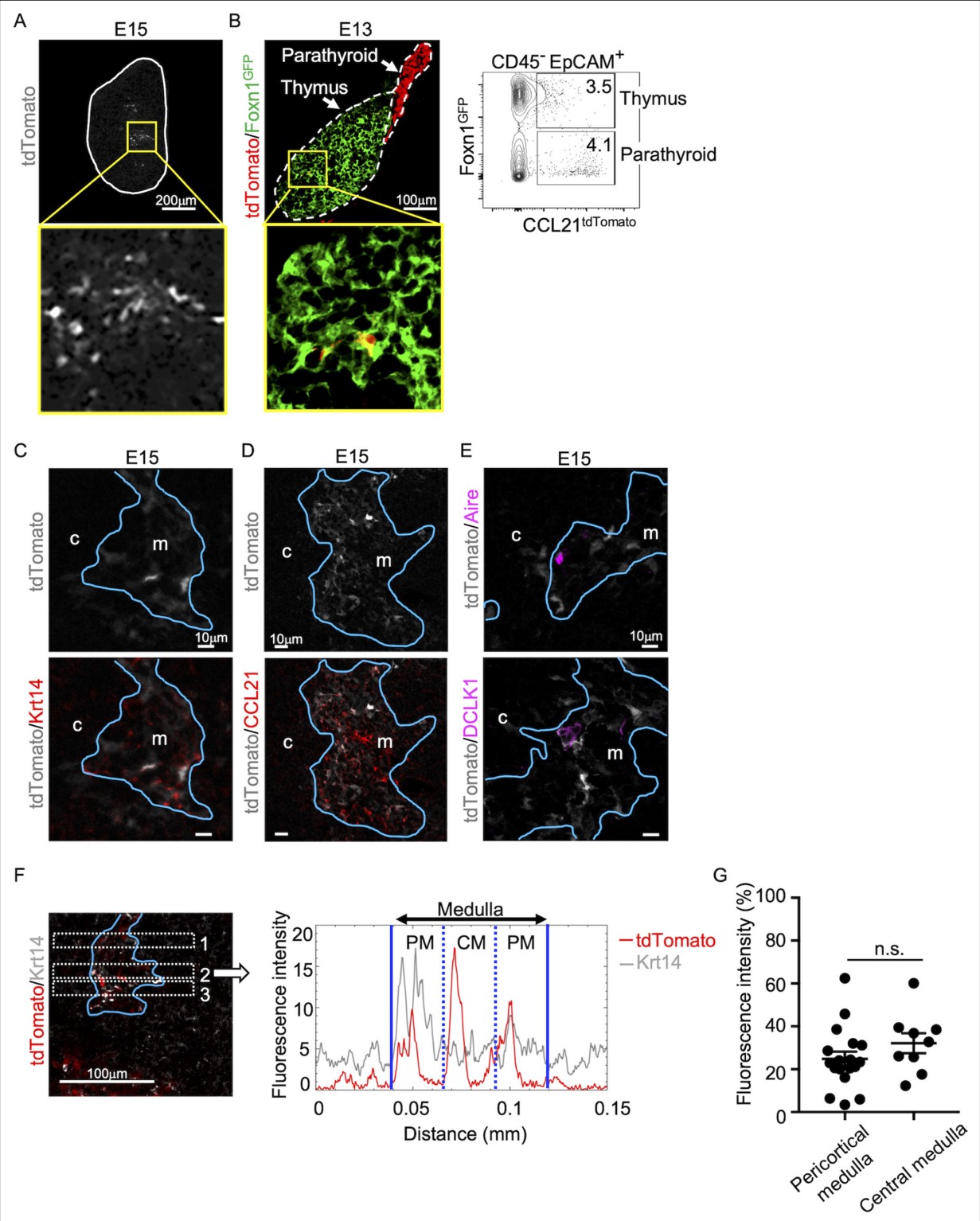

**Figure 2.** CCL21-expressing medullary thymic epithelial cells (mTECs) in embryonic thymus. (**A**) tdTomato fluorescence (white) detected in thymus sections from E15 *Ccl21a^tdTomato/+* embryos. White lines indicate the capsular outline of the thymus. The image in central region identified by the box in upper panel is magnified in bottom panel. Representative data from at least three independent experiments are shown. Scale bar, 200 µm. (**B**) tdTomato (red) and GFP (green fluorescence protein) (green) fluorescence in thymus sections from E13 *Ccl21a^tdTomato/+* × Foxn1-GFP-transgenic embryos (left). Dashed lines show the outlines for the thymus and parathyroid primordia. Central region of the thymus identified by the box in upper panel is magnified in bottom panel. Scale bar, 100 µm. Right panel shows a representative flow cytometric profile for tdTomato and GFP in CD45⁻EpCAM⁺ epithelial

*Figure 2 continued on next page*

*Figure 2 continued*

cells. Numbers in the contour plot indicate frequency of cells within indicated area. (**C–E**) Immunofluorescence analysis of thymus sections from E15 *Ccl21a^tdTomato/+^* embryos. tdTomato fluorescence (white) was detected with Krt14 (red) (**C**), CCL21 (red) (**D**), Aire, or DCLK1 (magenta) (**E**). Blue lines show the cortico-medullary junction. Representative data from at least three independent experiments are shown. c, cortex. m, medulla. Scale bar, 10 μm. (**F**) Distribution of tdTomato-expressing cells in the medullary region defined by Krt14 expression. Left panel shows representative tdTomato (red) and Krt14 (white) fluorescence signals in the thymus sections from *Ccl21a^tdTomato/+^* E15 embryos. Scale bar, 100 μm. Right panel shows fluorescence intensity profiles of tdTomato (red line) and Krt14 (gray line) signals within the region of interest (ROI) defined by dashed rectangles in the left panel. The medullary regions were equally divided into three areas into pericortical areas (PM) and central area (CM). (**G**) tdTomato intensity (means and standard error of the means [SEMs], *n* = 3) in indicated areas (**F**) and *Figure 1B* were calculated in comparison with total tdTomato intensity within the ROI. n.s., not significant.

The online version of this article includes the following figure supplement(s) for figure 2:

**Figure supplement 1.** Distribution of *Ccl21a*-expressing cells in embryonic thymus.

## Ccl21a-expressing mTECs arise early in central region of thymus primordium

It is further interesting to note that EGFP⁺ cells within Foxn1⁺ mTECs were detected in the central region of the thymus primordium in *Ccl21a*-Cre × CAG-loxP-EGFP mice as early as E11 (*Figure 5A*). The detection of EGFP⁺ cells occurred earlier in the ontogeny than the detection of Ccl21a^tdTomato+^ mTECs by E13 (*Figure 2B*) and CCL21-protein-expressing mTECs by E15 (*Figure 2D*). These sequential detections suggested that low-level *Ccl21a* transcription initiated as early as E11 during thymus organogenesis was sufficient to produce functional Cre proteins to mediate the expression of CAG-promoter-dependent EGFP proteins, but failed to produce a detectable amount of tdTomato or CCL21 proteins. Subsequent elevation of *Ccl21a* transcription level resulted in the production of detectable amounts of tdTomato proteins followed by CCL21 proteins. Indeed, single-cell RNA-sequencing analysis of *Ccl21a* transcript counts demonstrated that *Ccl21a* expression levels within individual *Epcam⁺ Foxn1⁺* TECs elevated during the embryogenesis from E12 to E14 (*Figure 5B*). At E14, *Ccl21a^low^* cells were detected at an equivalent frequency in β5t⁺ and β5t⁻ TEC populations (*Figure 5B*). *Ccl21a* transcription in CCL21-expressing mTECs was further elevated from embryonic to postnatal development, as shown by single-cell RNA-sequencing analysis of *Ccl21a* transcript counts (*Figure 5B*) and verified by flow cytometric analysis of *Ccl21a*-dependent tdTomato expression levels (*Figure 5C*). Importantly, *Ccl21a⁺Foxn1⁺* mTECs detected during early thymus organogenesis were localized in the central region of the thymus primordium and were anatomically and transcriptomically distinguishable from *Ccl21a⁺Gcm2⁺* parathyroid epithelial cells (*Figures 2B and 5A, D*). These results reveal that *Ccl21a⁺* mTECs develop in the central region of the thymus primordium as early as E11 before the generation of detectably CCL21 protein-expressing mTECs in the embryonic thymus by E15.

## CCL21-expressing mTECs are distinct from RANK-expressing mTECs

Along with the fate-mapping detection of the developmental potential in *Ccl21a⁺* cells, the early and temporal detection of *Ccl21a⁺* mTECs and CCL21-expressing mTECs during embryogenesis tempted us to speculate that *Ccl21a⁺* mTECs, and even CCL21-expressing mTECs, might possess mTEC progenitor activity. Previous studies reported that embryonic thymus develops RANK-expressing mTEC-restricted progenitors, which have a developmental capability to give rise to self-antigen-displaying mTECs, including Aire⁺ mTECs (*Baik et al., 2016*; *Akiyama et al., 2016*). RANK-expressing mTEC progenitors arise in the thymus during embryogenesis downstream of mTEC stem cells, which originate from cTEC-trait-expressing bipotent progenitors (*Takahama et al., 2017*; *Baik et al., 2016*; *Akiyama et al., 2016*; *Sekai et al., 2014*; *Ohigashi et al., 2015*). We next examined whether CCL21-expressing mTECs might overlap with RANK-expressing mTEC progenitors in *Ccl21a^tdTomato^ Tnfrsf11a(RANK)^Venus^* embryonic thymus. Interestingly, however, we found that CCL21- and RANK-expressing mTECs were largely distinct cell populations within the embryonic thymus (*Figure 6A, B*), although they were closely localized within the medullary region (*Figure 6A*). RNA-sequencing analysis of CCL21⁺RANK⁻ and CCL21⁻RANK⁺ TECs highly purified from *Ccl21a^tdTomato^ Tnfrsf11a(RANK)^Venus^* embryonic thymus (*Figure 6—figure supplement 1*) demonstrated that CCL21-expressing mTECs were markedly different from RANK-expressing mTECs in global gene expression profiles (*Figure 6C*). Gene ontology enrichment analysis ascertained the difference in global gene expression profiles in CCL21- and RANK-expressing mTECs (*Figure 6D*). Indeed, CCL21-expressing mTECs retained

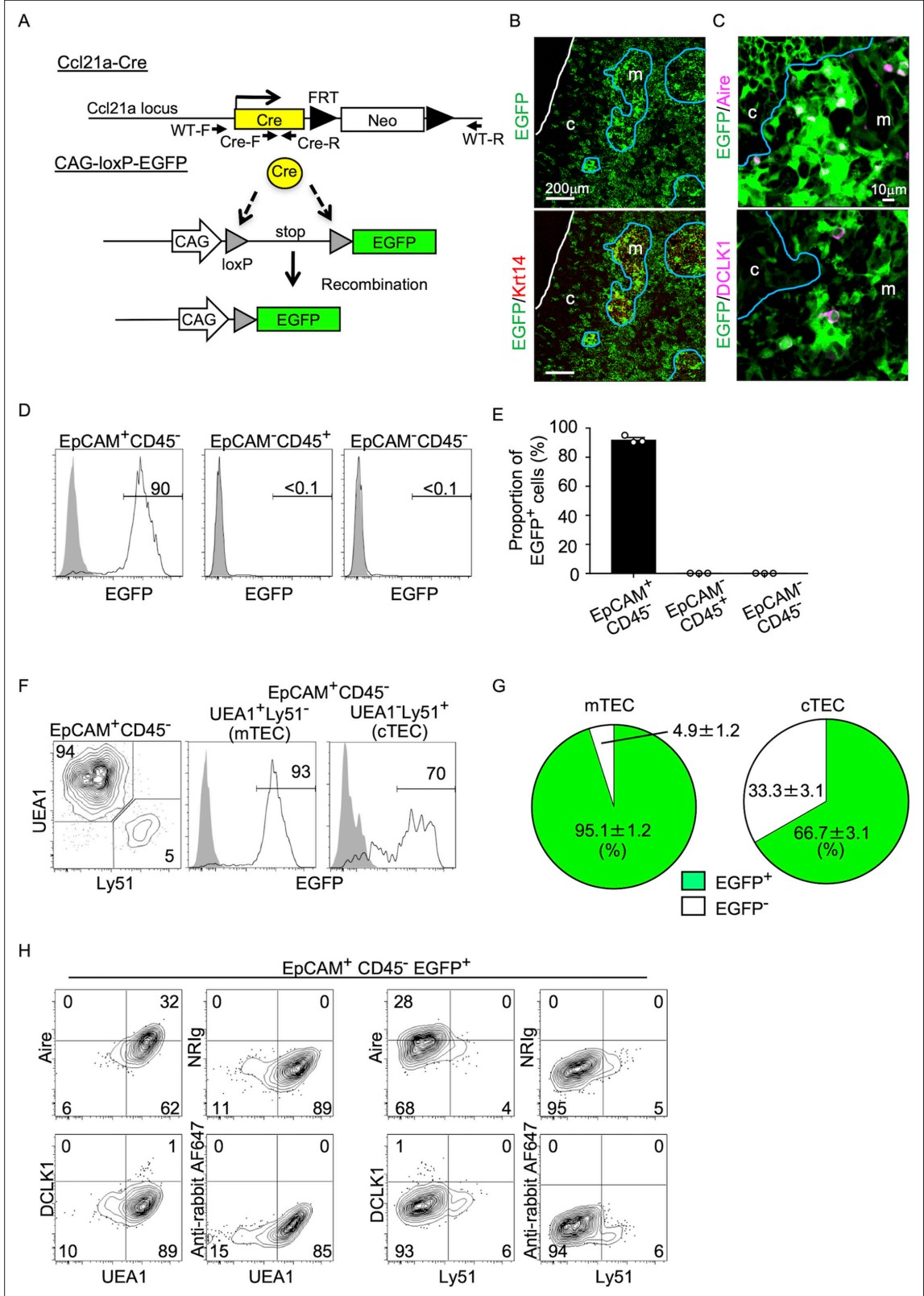

**Figure 3.** Fate mapping of *Ccl21a*-expressing cells in the thymus. (**A**) Schematic illustration of Cre-dependent *Ccl21a*-specific EGFP expression in *Ccl21a*-Cre × CAG-loxP-stop-loxP-EGFP mice. FRT, flippase recognition target site. Neo, neomycin resistance gene. Arrows indicate PCR primers for allele genotyping. (**B, C**) Immunofluorescence analysis of thymus sections from 3-week-old *Ccl21a*-Cre × CAG-loxP-EGFP mice. EGFP fluorescence (green) was detected with Krt14 (red) (**B**), Aire, or DCLK1 (magenta) (**C**). White lines indicate the capsular outline of the thymus. Blue lines show the

*Figure 3 continued on next page*

*Figure 3 continued*

cortico-medullary junction. Representative data from at least three independent experiments are shown. c, cortex. m, medulla. Scale bar, 200 μm (**B**) or 10 μm (**C**). (**D**) Flow cytometric analysis of liberase-digested thymus cells isolated from 3-week-old *Ccl21a*-Cre × CAG-loxP-EGFP mice. Histograms show EGFP fluorescence expression in EpCAM⁺CD45⁻ cells (left), EpCAM⁻CD45⁺ cells (middle), and EpCAM⁻CD45⁻ cells (right) prepared from *Ccl21a*-Cre × CAG-loxP-EGFP mice (black histograms) and littermate control (gray shaded histograms). Numbers in histograms indicate frequency of cells within indicated area. (**E**) Proportions (means and standard error of the means [SEMs], n = 3) of EGFP⁺ cells in indicated cell populations analyzed as in (**D**). (**F**) Flow cytometric profiles of UEA1 reactivity and Ly51 expression in EpCAM⁺CD45⁻ TECs (left). Histograms show EGFP fluorescence expression in EpCAM⁺CD45⁻UEA1⁺Ly51⁻ medullary thymic epithelial cells (mTECs; middle) and EpCAM⁺CD45⁻UEA1⁻Ly51⁺ cTECs (right) from *Ccl21a*-Cre × CAG-loxP-EGFP mice (black histograms) and littermate control (gray shaded histograms). Numbers in the histograms indicate frequency of cells within indicated area. (**G**) Proportions (means and SEMs, n = 3) of EGFP⁺ and EGFP⁻ cells in EpCAM⁺CD45⁻UEA1⁺Ly51⁻ mTECs (left) and EpCAM⁺CD45⁻UEA1⁻Ly51⁺ cTECs (right) analyzed as in (**F**). (**H**) Flow cytometric analysis of liberase-digested thymus cells isolated from *Ccl21a*-Cre × CAG-loxP-EGFP mice at 3 weeks old. Contour plots show Aire (top), DCLK1 (bottom), or isotype control detection with UEA1 reactivity or Ly51 expression in EpCAM⁺CD45⁻EGFP⁺ TECs. Numbers in the plots indicate frequency of cells within the indicated area. Representative data from three independent experiments are shown.

more cTEC-trait genes, including *Psmb11*, *Cxcl12*, *Cd83*, and *Ackr4*, than RANK-expressing mTECs (*Figure 6E*), suggesting that in comparison with RANK-expressing mTECs, CCL21-expressing mTECs are more recently derived from cTEC-trait-expressing bipotent TEC progenitors. However, similar to RANK-expressing mTECs, CCL21-expressing mTECs expressed equivalent levels of some mTEC-trait transcripts, including *Krt5* and *Krt14* (*Figure 6E*). Among the genes associated with TEC progenitor activity (*Senoo et al., 2007*; *Wallin et al., 1996*; *Ulyanchenko et al., 2016*; *Lucas et al., 2023*; *Hamazaki et al., 2007*; *Wong et al., 2014*), *Trp63* and *Pax1* were predominant in CCL21-expressing mTECs, whereas *Plet1*, *Krt19*, *Cldn3*, *Cldn4*, and *Ly6a* were readily detectable in RANK-expressing mTECs (*Figure 6E*). Thus, CCL21-expressing mTECs in embryonic thymus are largely distinct from RANK-expressing mTECs, whereas CCL21- and RANK-expressing mTECs are proximally localized with each other within the thymus medulla.

Nonetheless, it is important to point out that fate-mapping experiments using *Ccl21a*-Cre × CAG-loxP-EGFP mice were unable to distinguish whether TEC progenitor activity resides in CCL21-expressing mTECs generated within the thymus or in CCL21-expressing non-mTECs including neighboring parathyroid cells.

## Developmental potential of CCL21-expressing mTECs

To directly examine whether CCL21-expressing mTECs have a developmental potential to give rise to self-antigen-displaying mTECs, we finally traced the fate of CCL21-expressing mTECs within the thymus microenvironment. To do so, we employed reaggregate thymus organ culture experiments (*Anderson et al., 1993*; *White et al., 2008*) where RelB-deficient thymic stromal cells do not generate any functional mTECs (*Weih et al., 1995*; *Burkly et al., 1995*). Reaggregate thymuses were transplanted under mouse kidney capsules for reconstitution with hematopoietic cells derived from recipient mice. In this experimental condition, RelB-deficient thymic stromal cells were reaggregated with tdTomato-expressing mTECs highly purified from *Ccl21a^tdTomato* E17 embryos (*Figure 8—figure supplement 1*). Ccl21a^tdTomato+ mTECs had already become functional CCL21-protein-expressing mTECs by E17 (*Figure 2*), and the accumulation of CD4⁺CD8⁻ and CD4⁻CD8⁺ thymocytes in E17 thymic medullary region was defective in mice deficient in CCL21 (*Figure 7A–C*), so that the reaggregate thymus experiments would address whether functionally relevant CCL21-expressing mTECs give rise to other mTEC subpopulations including Aire⁺ mTECs.

We found that Ccl21a^tdTomato-positive EpCAM⁺CD45⁻ mTECs isolated from *Ccl21a^tdTomato/+* E17 embryos generated Aire⁺Krt14⁺ mTECs in the thymus reaggregated with RelB-deficient thymic stromal cells (*Figure 8A*). Aire⁺Krt14⁺ mTECs also developed from an equal number of parallelly isolated *Ccl21atdTomato*-negative EpCAM⁺CD45⁻ TECs (*Figure 8B*), which included cTEC–mTEC bipotent TEC progenitors (*Takahama et al., 2017*; *Nusser et al., 2022*)/. RelB-deficient thymic stromal reaggregates without additional TECs failed to generate Aire⁺Krt14⁺ mTECs (*Figure 8C*), whereas control organ culture using RelB-heterozygous thymus apparently developed medullary regions that contained Aire⁺Krt14⁺ mTECs (*Figure 8D*). Quantitative evaluation of Krt14⁺ medullary areas (*Figure 8E*) within total thymus aggregates (*Figure 8F*) demonstrated that *Ccl21a*⁺ mTECs have a developmental potential to give rise to Aire⁺Krt14⁺ mTECs, equivalently to and not statistically different from *Ccl21a*-negative TECs, including bipotent TEC progenitors (*Figure 8G, H*). Interestingly, *Ccl21a*⁺ mTECs were

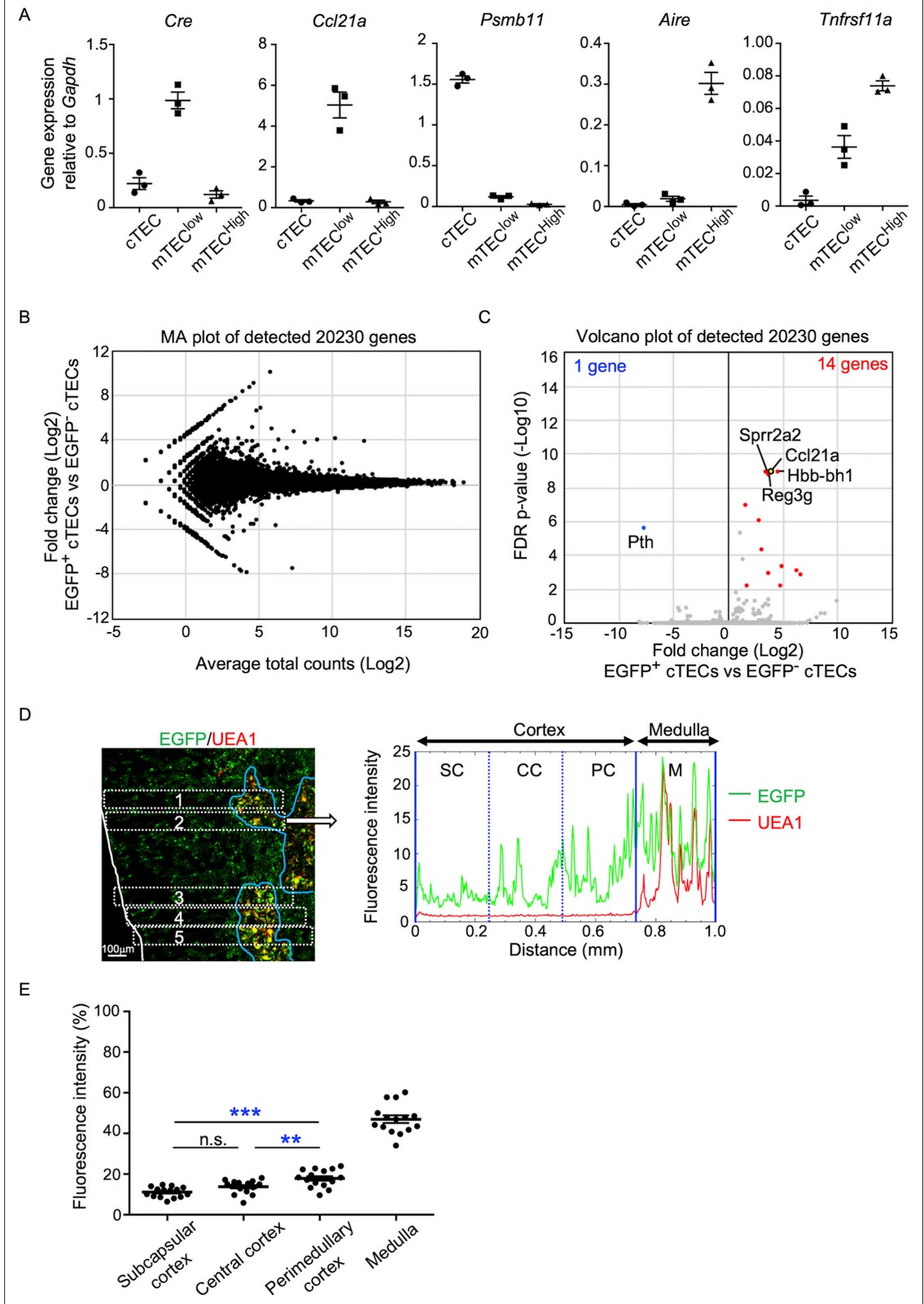

**Figure 4.** Characterization of cTECs that previously transcribed *Ccl21a*. (**A**) Quantitative RT-PCR analysis of *Cre* and indicated genes (means and standard error of the means [SEMs], *n* = 3) in cTECs (EpCAM⁺CD45⁻UEA1⁻Ly51⁺), mTEC^low (EpCAM⁺CD45⁻UEA1⁺Ly51⁻ I-A^low), and mTEC^high (EpCAM⁺CD45⁻UEA1⁺Ly51⁻ I-A^high) isolated from *Ccl21a^{Cre/+}* mice. (**B**) MA plot of 20,230 genes detected in RNA-sequencing analysis of EGFP⁺ and EGFP⁻ cTECs isolated from *Ccl21a*-Cre × CAG-loxP-EGFP mice at 2-week-old. Detected genes are plotted as log₂ average total counts versus log₂ fold

*Figure 4 continued on next page*

*Figure 4 continued*

changes (EGFP$^+$ cTECs/EGFP$^-$ cTECs). (**C**) Volcano plot analysis of EGFP$^+$ and EGFP$^-$ cTECs. Detected genes are plotted as log$_2$ fold changes (EGFP$^+$ cTECs/EGFP$^-$ cTECs) versus $-$log$_{10}$ FDR p-values. Fourteen genes (red symbols) are more highly detected (log$_2$ fold change >1.7, FDR p-value <0.05) in EGFP$^+$ cTECs than EGFP$^-$ cTECs, whereas one gene (blue symbol) is more highly detected (log$_2$ fold change <$-$1.7, FDR p-value <0.05) in EGFP$^-$ cTECs than EGFP$^+$ cTECs. (**D**) Distribution of EGFP-expressing cells in the thymus. Left panel shows representative fluorescence signals for EGFP (green) in the thymus sections from *Ccl21a*-Cre × CAG-loxP-EGFP mice. The medullary region is defined by UEA1 reactivity (red). White lines indicate the capsular outline of the thymus. Blue lines show the cortico-medullary junction. Scale bar, 100 µm. Right panel shows fluorescence intensity profiles of EGFP (green line) and UEA1 (gray line) signals within the region of interest (ROI) defined by dashed rectangles in the left panel. Cortical regions were equally divided into three areas into subcapsular cortex (SC), central cortex (CC), and perimedullary cortex (PC). M, medulla. (**E**) EGFP intensity (means and SEMs, *n* = 3) in indicated areas (**C**) and *Figure 1C* were calculated in comparison with total EGFP intensity within the ROI. n.s., not significant. **p < 0.01, ***p < 0.001.

The online version of this article includes the following figure supplement(s) for figure 4:

**Figure supplement 1.** Purity of isolated EGFP$^+$ and EGFP$^-$ cTECs for transcriptomic analyses.

**Figure supplement 2.** Transcriptomic profiles of cTECs, medullary thymic epithelial cells (mTECs), EGFP$^+$ cTECs, and EGFP$^-$ cTECs.

**Figure supplement 3.** Distribution of EGFP$^+$ cells in the thymus.

still detected in the Krt14$^+$ medullary areas in reconstituted thymus reaggregated with either *Ccl21a*$^+$ mTECs or *Ccl21a*-negative TECs (*Figure 8I, J*), indicating that *Ccl21a*$^+$ mTECs have a potential to maintain themselves in the thymus and that *Ccl21a*-negative TECs have a developmental potential to produce *Ccl21a*$^+$ mTECs. Most importantly, these results directly indicate that functionally relevant CCL21-expressing mTECs have a developmental potential to give rise to Aire-expressing mTECs.

In contrast to CCL21-expressing mTECs isolated from E17 embryos, Ccl21a$^{tdTomato}$-positive EpCAM$^+$CD45$^-$ mTECs isolated from 4-week-old *Ccl21a*$^{tdTomato/+}$ mice did not exhibit any signs of Aire$^+$Krt14$^+$ mTEC generation in the thymus reaggregated with RelB-deficient thymic stromal cells (*Figure 8—figure supplement 2A–G*), suggesting a decline in the developmental potential of CCL21$^+$ mTECs during the ontogeny from embryonic to postnatal period. However, no detection of progeny could also result from technical caveats associated with reaggregate thymus organ culture conditions that were suitable for detection of the developmental potential of embryonic but not postnatal CCL21-expressing mTECs, or differences in the lifespan and/or turnover of embryonic and postnatal CCL21-expressing cells.

Throughout the development including E17 embryos, CCL21-expressing mTECs were localized specifically in the central region of the thymus and were Krt14$^+$ (*Figures 1, 2, 6, and 9A*). However, single-cell RNA-sequencing analysis demonstrated that the global gene expression profiles were largely different between embryonic *Ccl21a*$^{low}$ mTECs and 4-week-old postnatal *Ccl21a*$^{high}$ mTECs (*Figure 9B*). Indeed, the majority of CCL21-expressing mTECs at E17 but not the postnatal period simultaneously expressed cTEC-associated protein Ly51 and mTEC-associated UEA1-binding molecules on the cell surface (*Figure 9C*). Ly51 expression was lower in embryonic TECs than postnatal TECs (*Ripen et al., 2011*) and equivalently distributed in the thymus including the medullary region defined by Krt14-expressing mTECs centrally localized in E17 thymus (*Figure 9A*). These results highlight the difference in molecular expression profiles, including Ly51 expression, between E17 CCL21- and postnatal CCL21-expressing mTECs.

## Discussion

The present results demonstrate that CCL21-expressing mTECs arise early during embryonic thymus development. *Ccl21a*-Cre-mediated detection reveals that *Ccl21a*$^+$*Foxn1*$^+$ mTECs are detectable as early as E11 in the central region of the embryonic thymus, shortly after the organogenesis of the thymus primordium, and this is followed by the detection of *Ccl21a-tdTomato*-expressing mTECs by E13 and CCL21 protein-expressing mTECs by E15. *Ccl21a*-Cre-mediated fate-mapping analysis establishes that at 3 weeks old, a vast majority of mTECs, including the self-antigen-displaying mTECs such as Aire$^+$ mTECs and thymic tuft cells, and approximately two-thirds of cTECs originate from *Ccl21a*$^+$ cells. Most importantly, our results demonstrate that *Ccl21a-tdTomato*-expressing and functional CCL21 protein-producing mTECs isolated from E17 embryonic thymus have a developmental potential to give rise to the self-antigen-displaying mTECs in the thymus microenvironment. These results demonstrate a previously unknown process by which the diversity in the thymus medulla epithelium

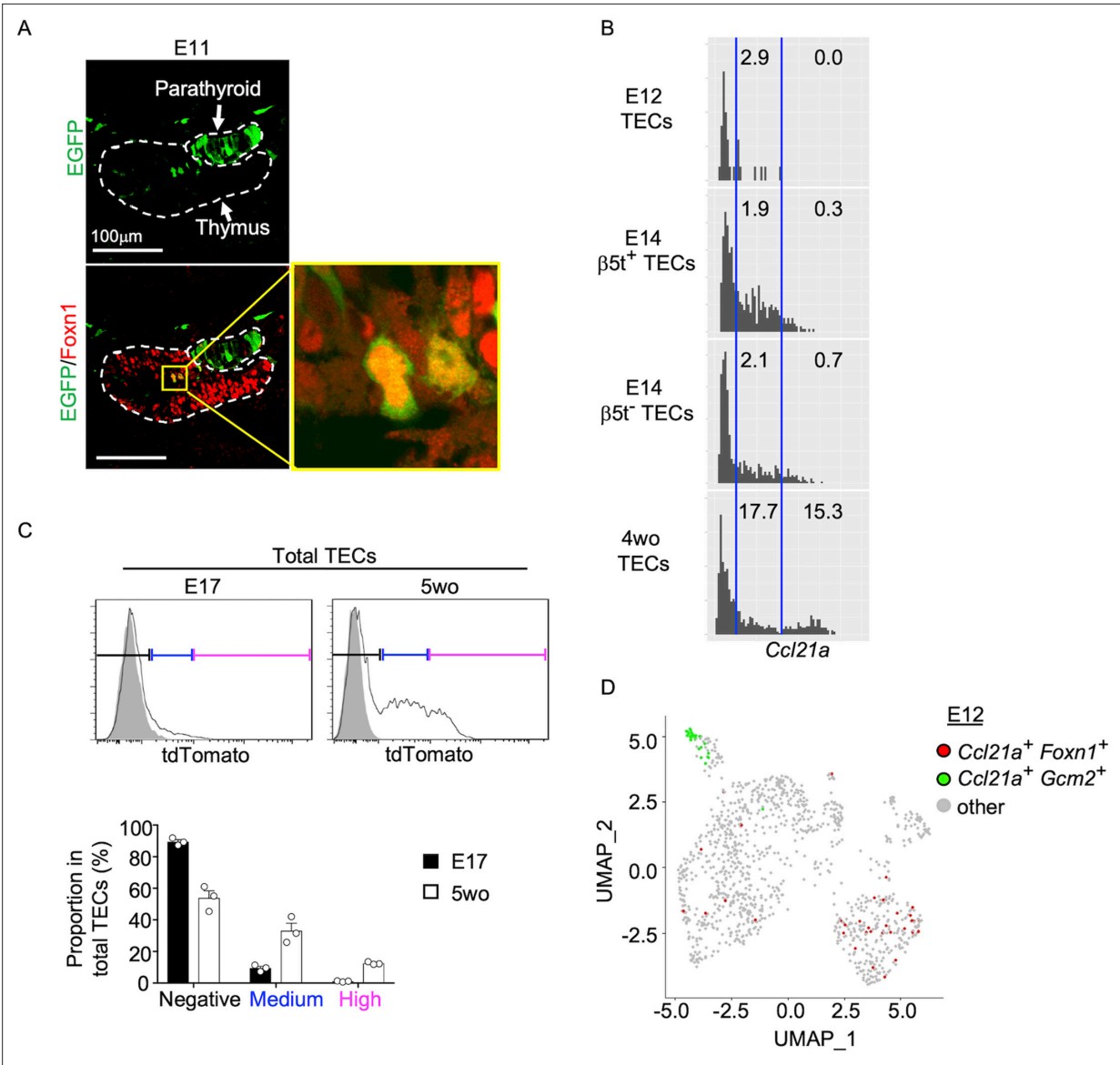

**Figure 5.** *Ccl21a*-expressing medullary thymic epithelial cells (mTECs) during early thymus organogenesis. (**A**) Immunofluorescence analysis of thymus section from *Ccl21a*-Cre × CAG-loxP-EGFP E11 embryos. EGFP (green) and Foxn1 (red) were analyzed as indicated. Dashed lines show the outline of the thymus and parathyroid primordia. Image in yellow box in the middle of the thymus is magnified in right panel. Scale bar, 100 μm. Representative data from two independent experiments are shown. (**B**) Single-cell RNA-sequencing analysis of *Epcam*+*Foxn1*+ TECs from E12, E14, and 4-week-old β5t-Venus knock-in mice. Normalized log transcript counts of *Ccl21a* mRNA (*x*-axis) and cell numbers (*y*-axis) are plotted. For E14 TECs, β5t-Venus+ and β5t-Venus− TECs were sorted and analyzed in parallel. Numbers in histograms indicate frequency of cells within indicated area. (**C**) Flow cytometric analysis of tdTomato expression in EpCAM+CD45− TECs isolated from E17 embryonic and 2-week-old postnatal *Ccl21a^{tdTomato/+}* mice (black histograms) and WT mice (gray shaded histograms). Bottom plots indicate the proportions (means and standard error of the means [SEMs], *n* = 3) of tdTomato^{negative} (black), tdTomato^{medium} (blue), and tdTomato^{high} (magenta) cells within EpCAM+CD45− TECs as defined in top histograms. (**D**) Single-cell RNA-sequencing analysis of *Epcam*+ E12 embryonic pharyngeal epithelial cells. Dots indicate Uniform Manifold Approximation and Projection (UMAP) plot of *Ccl21a*+*Foxn1*+ mTECs (red), *Ccl21a*+*Gcm2*+ parathyroid epithelial cells (green), and other cells (gray).

is generated during embryogenesis by the developmental conversion of the CCL21-expressing thymocyte-attracting mTEC subset into the self-antigen-displaying mTEC subset, including Aire-expressing mTECs and thymic tuft cells.

Functional conversion of the thymocyte-attracting mTEC subset into the self-antigen-displaying mTEC subset offers an interesting implication that the thymus medulla is formed by initially generating CCL21-expressing mTECs and attracting CCR7-expressing positively selected thymocytes from the cortex even before the subsequent display of self-antigens mediated by Aire+ mTECs and thymic

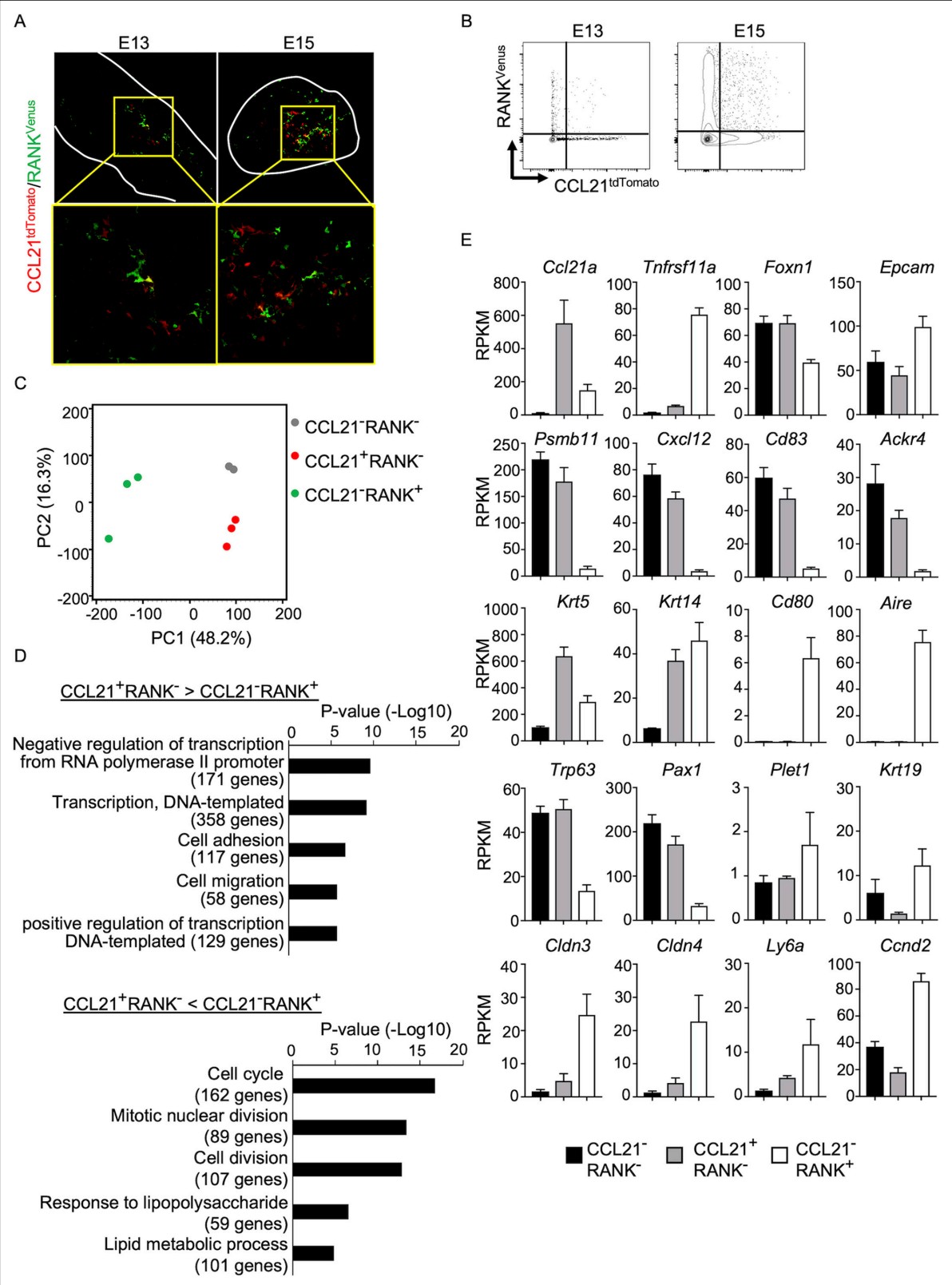

**Figure 6.** CCL21-expressing medullary thymic epithelial cells (mTECs) are distinct from RANK-expressing mTECs. (**A**) Immunofluorescence analysis of tdTomato (red) and Venus (green) signals in thymus sections from *Ccl21a^tdTomato/+Tnfrsf11a(RANK)^Venus* mice at indicated embryonic age. White lines indicate the capsular outline of the thymus. Images in boxed regions in upper panels are magnified in bottom panels. Representative data from at least three independent experiments are shown. (**B**) Flow cytometric profiles of tdTomato and Venus fluorescence signals in EpCAM⁺CD45⁻ TECs isolated

*Figure 6 continued on next page*

*Figure 6 continued*

from *Ccl21a*<sup>tdTomato</sup> *Tnfrsf11a(RANK)*<sup>Venus</sup> mice at indicated embryonic age. (**C**) Principal component analysis of RNA-sequencing data of indicated cell populations isolated from *Ccl21a*<sup>tdTomato/+</sup>*Tnfrsf11a(RANK)*<sup>Venus</sup> E17 embryos. (**D**) Enrichment analysis of ontology for genes that are differently expressed (RPKM >1, fold change >1.5 or <—1.5, FDR p-value <0.05) between CCL21⁺RANK⁻ and CCL21⁻RANK⁺ TECs. Bars show the adjusted p-values of the top five categories enriched in CCL21⁺RANK⁻ TECs (top) and CCL21⁻RANK⁺ TECs (bottom). Numbers in parentheses indicate the number of categorized genes. (**E**) RPKM values of indicated genes detected in RNA-sequencing analysis.

The online version of this article includes the following figure supplement(s) for figure 6:

**Figure supplement 1.** Purity of isolated TECs for transcriptomic analysis.

mimetic cells. Failure to establish self-tolerance by the loss of *Ccr7* or *Ccl21a* in the thymus (*Kozai et al., 2017*; *Kurobe et al., 2006*; *Nitta et al., 2009*) underscored the importance of functional CCL21-expressing mTEC subset. Our present results showing developmental potential of functional CCL21-expressing mTECs to give rise to self-antigen-displaying mTECs reveals an additionally important function of CCL21-expressing mTECs in forming the diversity in thymus medulla epithelium, including the self-antigen-displaying mTEC subset. Stepwise progression in mTEC development from thymocyte-attracting function to self-antigen-displaying function fits stepwise requirement to initially recruit newly generated T cells from the thymic cortex before tolerizing T cells to self-antigens. Downregulation of CCL21 in functionally converted self-antigen-displaying mTECs may further contribute to the prevention of excessive negative selection of newly generated T cells to self-antigens.

Recent studies demonstrated that the function of the thymus medulla is not limited to the establishment of self-tolerance in conventional αβ T cells but extends to the development of innate lymphocytes, including invariant NKT cells and γδ T cells (*White et al., 2014*; *Roberts et al., 2012*). The developmental capability of CCL21-expressing mTECs may further extend to additional mTEC subpopulations that contribute to supporting the development of innate invariant lymphocytes. Thus, CCL21-expressing mTECs may represent a primordial mTECs which separate newly generated αβ T cells, γδ T cells, and NKT cells from the thymic cortex to the medullary microenvironment for further selection and maturation.

In contrast to the robust generation of Aire⁺ mTECs from embryonic CCL21⁺ mTECs, our results show that the ability to give rise to Aire⁺ mTECs was not detectable in postnatal CCL21⁺ mTECs. The fact that no Aire⁺ mTECs were detected in reaggregate thymus organ culture may reflect a postnatal

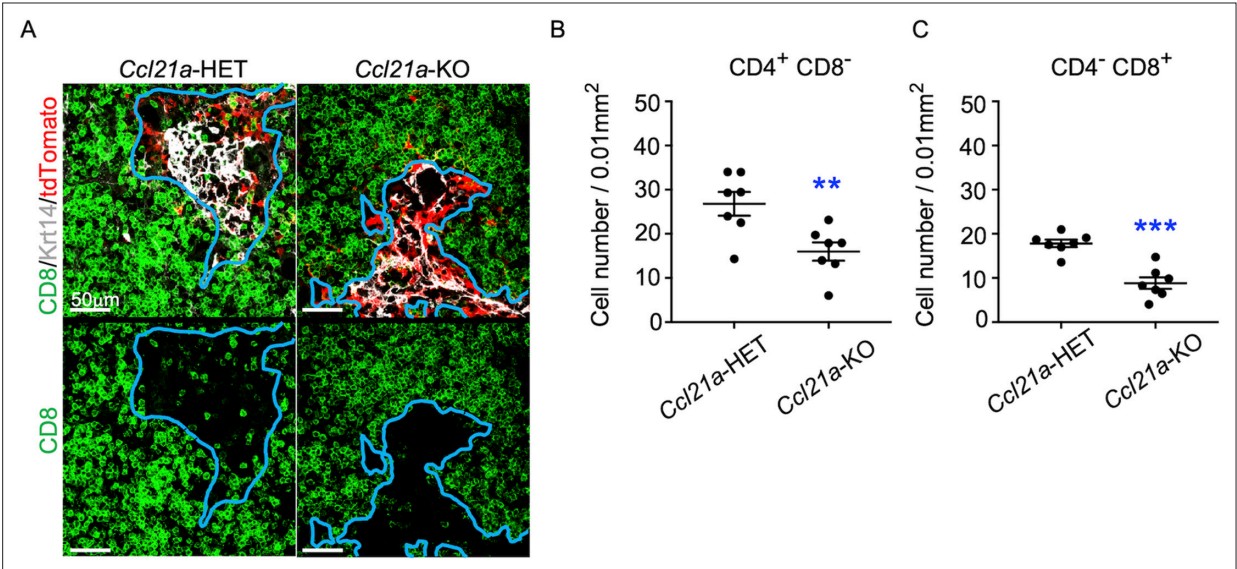

**Figure 7.** CCL21-expressing medullary thymic epithelial cells (mTECs) in E17 embryos are functional in thymocyte attraction. (**A**) Immunofluorescence analysis of CD4, CD8, Krt14, and tdTomato in thymus sections from heterozygous (*Ccl21a*<sup>tdTomato/+</sup>) control mice or homozygous (*Ccl21a*<sup>tdTomato/tdTomato</sup>) CCL21-deficient mice at E17. E17 thymus sections were stained for CD4, CD8, and Krt14. Representative images for Krt14 (white), tdTomato fluorescence (red), and CD8 (green) expressions are shown. Blue lines indicate cortico-medullary junctions, according to the distribution of Krt14-expressing mTECs. tdTomato-positive cells were localized in the medullary regions. Scale bar, 50 µm. (**B, C**) Graphs show the means and standard error of the means (SEMs; n = 3) of the numbers of CD4⁺CD8⁻ (**B**) and CD4⁻CD8⁺ (**C**) thymocytes per unit area (0.01 mm²) in the medullary regions. **p < 0.01; ***p < 0.001.

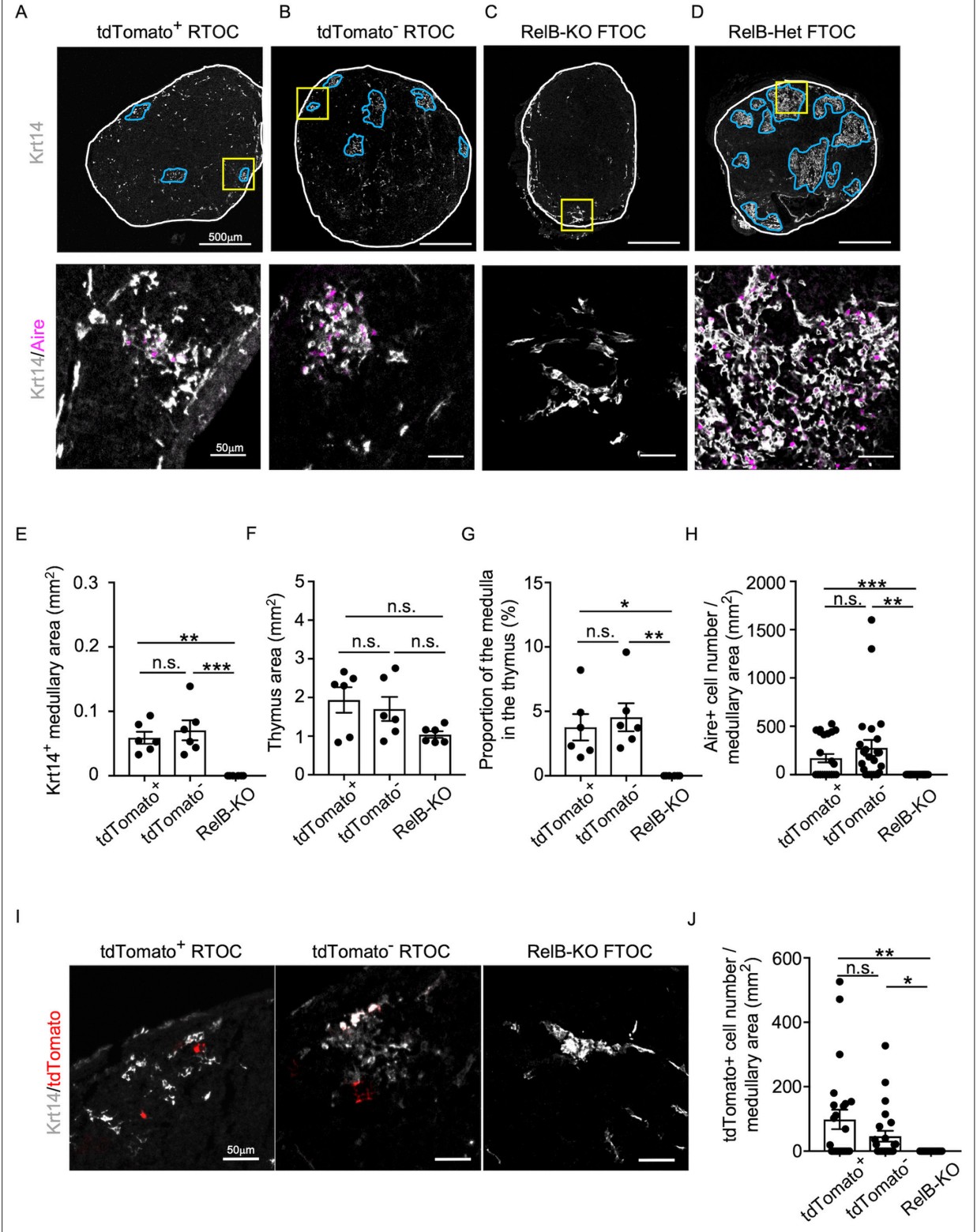

**Figure 8.** Developmental potential of CCL21-expressing medullary thymic epithelial cells (mTECs). (**A–D**) Immunofluorescent staining of indicated thymus graft sections. Shown in upper panels are Krt14 expression (white). White lines indicate capsular outline of the thymus. Blue lines show Krt14+ medullary region. Scale bar, 500 µm. Bottom panels show Krt14 (white) and Aire (magenta) fluorescence signals within yellow boxes in upper panels. Scale bar, 50 µm. Grafts represent RelB-KO thymic stroma reaggregated with tdTomato+ TECs (**A**) and tdTomato− TECs (**B**) as well as fetal thymus lobe from RelB-KO mice (**C**) and RelB-heterozygous mice (**D**). (**E**) Size of Krt14+ medullary areas (means and standard error of the means [SEMs], *n* = 6) in

*Figure 8 continued on next page*

*Figure 8 continued*

indicated thymus graft sections. (**F**) Size of grafted thymus areas (means and SEMs, *n* = 6) in indicated thymus graft sections. (**G**) Proportion (means and SEMs, *n* = 6) of Krt14+ medullary areas in the thymus areas in indicated thymus graft sections. (**H**) Numbers of Aire+ mTECs per mm² of Krt14+ medullary areas in indicated thymus graft sections. (**I**) Immunofluorescence analysis of tdTomato (red) and Krt14 (white) in indicated thymus graft sections. Scale bar, 50 μm. (**J**) Numbers of tdTomato+ cells per mm² of Krt14+ medullary areas in indicated thymus graft sections. All images are representative data from three independent experiments. *p < 0.05, **p < 0.01, ***p < 0.001, n.s., not significant.

The online version of this article includes the following figure supplement(s) for figure 8:

**Figure supplement 1.** Purity of isolated TECs for reaggregation with RelB-KO thymus stroma.

**Figure supplement 2.** Developmental potential of postnatal CCL21-expressing medullary thymic epithelial cells (mTECs).

decline in the developmental potential of CCL21+ mTECs. However, it is also possible that the developmental kinetics and the requirements for postnatal CCL21+ mTECs may be different from those for E17 CCL21+ mTECs, so that the reaggregate thymus organ culture conditions for detecting the development of E17 CCL21+ mTECs may not be optimal for the detection of the survival and development of postnatal CCL21+ mTECs.

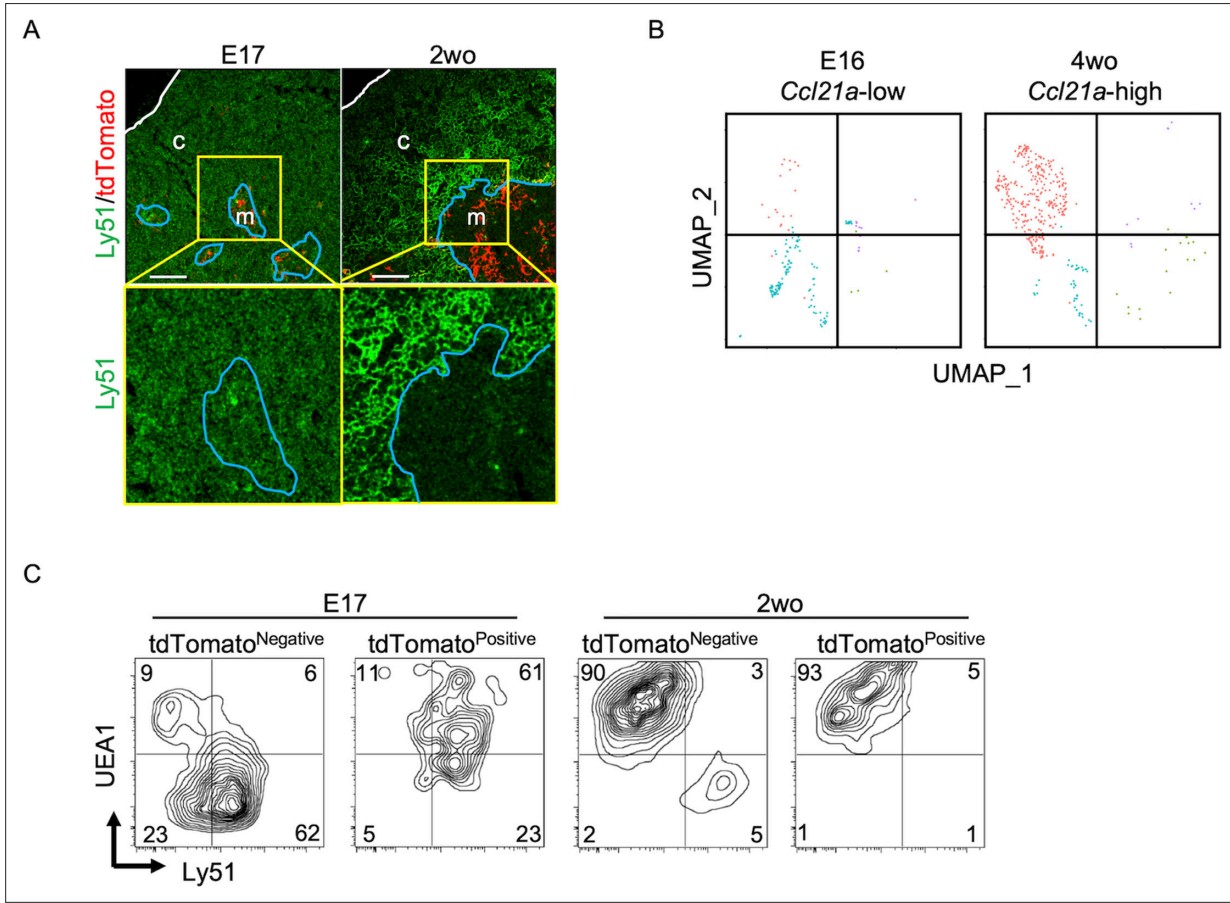

**Figure 9.** CCL21-expressing medullary thymic epithelial cells (mTECs) at E17 embryonic and postnatal period. (**A**) Immunofluorescence analysis of thymus sections from E17 or 2-week-old *Ccl21a^tdTomato/+* mice. tdTomato fluorescence (red) and Ly51 (green) expression are shown. White lines indicate thymic capsules. Blue lines indicate cortico-medullary junctions. Images in yellow boxes (top panels) are magnified in bottom panels. Representative data from at least three independent experiments are shown. c, cortex. m, medulla. Scale bar, 100 μm. (**B**) Uniform Manifold Approximation and Projection (UMAP) plots showing the clusters of *Ccl21a+* mTECs in single-cell RNA-sequencing analysis of thymic epithelial cells isolated from E16 embryonic and 4-week-old postnatal mice. The majority of E16 *Ccl21a+* mTECs were *Ccl21a^low*, whereas the majority of postnatal *Ccl21a+* mTECs were *Ccl21a^high* (**Figure 5B, C**). (**C**) Flow cytometric profiles of UEA1 reactivity and Ly51 expression in tdTomato^negative EpCAM+CD45− TECs and tdTomato^positive EpCAM+CD45− TECs isolated from E17 embryonic and 2-week-old postnatal *Ccl21a^tdTomato/+* mice. Numbers in the contour plots indicate frequency of cells within indicated area. Representative data from three independent experiments are shown.

Nevertheless, it is important to notice the difference in the developmental potential between embryonic and postnatal CCL21[+] mTECs in reaggregate thymus organ culture. Analysis of molecular expression profiles further highlighted the difference between embryonic and postnatal CCL21[+] mTECs. We think that CCL21[+] mTECs may be heterogenous in developmental capability. Embryonic CCL21[+] mTECs, which give rise to self-antigen-displaying mTECs including Aire[+] mTECs and mimetic mTECs, may also terminally differentiate into a subpopulation of CCL21[+] mTECs, which no longer have further developmental potential, and which represent the majority of postnatal CCL21[+] mTECs.

Interestingly, our results also reveal that approximately two-thirds of cTECs are derived from progenitor cells that previously transcribed *Ccl21a*. The previous expression of *Ccl21a* transcripts in the fraction of cTECs is influenced by the intrathymic proximity to the medullary region and is enriched in the perimedullary cortical region (*Figure 4E*). Developmental capability to become cTECs is shown in *Ccl21a* gene-expressing cells, and it is unknown whether CCL21-protein-expressing functional mTECs retain the developmental potential to give rise to cTECs. In contrast, our results from reaggregate thymus experiments clarified that CCL21-protein-expressing functional mTECs have a developmental potential to convert into Aire-expressing mTECs. Consequently, we think that at least a fraction of *Ccl21a* transcript-expressing mTECs retain a bipotent developmental potential to give rise to both cTECs and mTECs. It is unclear whether most *Ccl21a*-expressing mTECs retain a bipotent developmental potential for cTECs and mTECs or *Ccl21a*-expressing mTECs represent a mixture of bipotent progenitors and lineage-committed mTEC progenitors. Because of the unequal distribution of EGFP-labeled cTECs within the thymic cortex, initial signals to promote bipotent progenitors to mTEC lineage may be provided by molecules restricted in the central region of the embryonic thymus, and those signals may promote the transcription of *Ccl21a* at an early stage of mTECs, which still retain the potential to become cTECs. Further examination of cTEC potential, for example, by using CCL21-tdTomato-expressing cells that constitutively express GFP, may lead to a better understanding of the cTEC-generating capability in embryonic and postnatal CCL21[+] mTECs. It is also interesting to examine whether GFP[+] cTEC development in *Ccl21a*-Cre × CAG-loxP-EGFP mice is mediated through RelB-dependent mTEC developmental progression and/or dependent on developing thymocyte-dependent mTEC-nurturing 'crosstalk' signals.

*Nusser et al., 2022* recently identified two bipotent TEC progenitor populations: an early cTEC-biased progenitor population and a postnatal mTEC-biased progenitor population. Interestingly, they noted that the postnatally appearing mTEC-biased progenitor population includes cells that transcribe *Ccl21a* (*Nusser et al., 2022*). It would be interesting to clarify whether and how those postnatal TEC progenitors overlap with the embryonic and postnatal CCL21-protein-expressing mTECs reported in this study. It would also be interesting to shed light on how differently *Ccl21a*[+] progenitors contribute to cTECs and mTECs over the ontogeny and whether the enrichment of *Ccl21a*[+] progenitor-derived cTECs in the perimedullary area reflects a temporal replacement of cTECs derived from *Ccl21a*[+] progenitors localized in the medulla.

Further characterization of CCL21-expressing mTECs carrying a developmental potential to give rise to self-antigen-displaying mTEC subset is of great interest. Our results highlight the difference between CCL21-expressing mTECs and previously characterized RANK-expressing mTEC-restricted progenitors. CCL21- and RANK-expressing mTECs are detectable in E13 embryonic thymus, are proximally localized within the medullary region in the thymus, and are capable of generating Aire-expressing mTECs. Unlike RANK-expressing mTECs, however, CCL21-expressing mTECs include developmental potential to give rise to cTECs. Gene expression profiles show that CCL21-expressing mTECs retained more cTEC-trait genes, including *Psmb11*, *Cxcl12*, *CD83*, and *Ackr4*, than RANK-expressing mTECs, suggesting that in comparison with RANK-expressing mTECs, CCL21-expressing mTECs are more closely related to and more recently derived from cTEC-trait-expressing bipotent TEC progenitors. Furthermore, it is interesting to note that recently described Krt19[+] progenitors that give rise to multiple mature mTEC populations including Aire[+] mTECs, mimetic mTECs, and postnatal CCL21[+] mTECs, lack expression of CCL21 and RANK (*Lucas et al., 2023*). An important aspect of future work will be to examine whether CCL21-, RANK-, and Krt19-expressing mTECs are linked in upstream–downstream relationships in a single developmental pathway, or are developmentally distinct and parallel progenitors to give rise to mTECs. Similarly, it will be interesting to better clarify possible heterogeneity in CCL21-expressing mTECs in relation to their ability to give rise to cTECs as

well as mTECs. Nevertheless, the finding in our study that CCL21[+] embryonic mTECs can give rise to Aire[+] mTECs reveals a new source of an important component of the thymus medulla.

Our results show that 95% of mTECs are labeled with EGFP fluorescence and therefore derived from *Ccl21a*[+]progenitors, whereas 5% of mTECs are not labeled with EGFP fluorescence in *Ccl21a*-Cre × loxP-EGFP mice. Our previous study using β5t-driven Cre mice indicated fate-mapping of 95% EGFP[+] mTECs in CAG-loxP-stop-loxP-EGFP-transgenic reporter mice used in this study and >99.5% ZsGreen[+] mTECs in Rosa26-loxP-ZsGreen-knock-in reporter mice (*Ohigashi et al., 2013*). Consequently, we think that 95% EGFP[+] cells in mTECs detected in this study may indicate that virtually all (>99.5%) mTECs are derived from cells that once transcribed *Ccl21a*.

A previous study detected *Ccl21a* transcripts within a podoplanin-expressing TEC subpopulation that resembled immature TECs in postnatal thymus (*Onder et al., 2015*). It was reported that podoplanin-expressing mTEC progenitors, also known as junctional TECs (jTECs), were localized at the cortico-medullary junction (*Onder et al., 2015*). However, our results show that CCL21-expressing mTECs in the embryonic and postnatal thymus distribute throughout the medullary region and are not enriched in the cortico-medullary junctions (*Figures 1H and 2G*). Indeed, the majority of CCL21-expressing mTECs in postnatal thymus did not express podoplanin (*Figure 8—figure supplement 2H*). Moreover, podoplanin expression was broadly detected by the majority of embryonic E17 TECs, most of which were cTECs (*Figure 8—figure supplement 2H*). Thus, neither podoplanin expression nor cortico-medullary junctional distribution characterizes CCL21-expressing mTECs. More importantly, our results demonstrate developmental potential in thymocyte-attracting CCL21-expressing mTECs and identify developmental conversion between functionally distinct mTEC subsets, which serves to assemble a diverse medullary microenvironment in the thymus.

In conclusion, the present study establishes that CCL21-expressing functional mTEC subset carries a developmental potential to give rise to self-antigen-displaying mTECs, revealing that the sequential conversion of functionally distinct mTEC subsets from thymocyte-attracting cells into self-antigen-displaying cells contributes to the functional diversification of thymus medulla epithelium during embryonic development. The present results are likely to ignite many interesting questions, including the characterization of signals that drive the developmental conversion of mTECs and the elucidation of developmental pathways for diverse TEC subpopulations.

## Materials and methods

### Mice

C57BL/6 (B6) mice were obtained from The Jackson Laboratory (Bar Harbor, ME) and SLC (Shizuoka, Japan). *Ccl21a*-tdTomato knock-in mice (*Kozai et al., 2017*), *Relb*-knockout mice (*Burkly et al., 1995*), *Tnfrsf11a(RANK)*-Venus transgenic mice (*McCarthy et al., 2015*), *Psmb11(β5t)*-Venus knock-in mice (*Murata et al., 2007*), CAG-loxP-stop-loxP-EGFP transgenic mice (*Kawamoto et al., 2000*), and *Foxn1*-GFP transgenic mice (*Terszowski et al., 2006*) were described previously. All mouse experiments were performed with consent from the Animal Experimentation Committee of the University of Tokushima (T2022-50), the Animal Care and Use Committee of the National Cancer Institute (ASP 21-431, ASP 21-432, and EIB-076-3), the Birmingham Animal Welfare and Ethical Review Board and UK Home Office Animal Licences (PP7518148 and PP2990911), and the Institutional Animal Care and Use Committee of RIKEN Kobe Branch (A2001-03).

### Generation of Ccl21a-Cre mice

The targeting vector was prepared by subcloning *Ccl21a*-containing mouse genomic BAC fragments (Advanced GenoTechs) and Cre-encoding cDNA into a plasmid containing a pgk-neo cassette (Gene Bridges). The linearized targeting vector was introduced into TT2 ES cells (*Yagi et al., 1993*). Targeted knock-in alleles were screened by genomic PCR analysis and Southern blot analysis. The *Ccl21a*-Cre mice (https://large.riken.jp/distribution/mutant-list.html; Accession No. CDB1131K) are available to the scientific community and were backcrossed to B6 mouse strain for more than five generations. The knock-in allele is detected using PCR primers as follows: Cre-F, 5′-AGGTTCGTTCACTCATGGA-3′ and Cre-R, 5′-TCGACCAGTTTAGTTACCC-3′ (product size, 235 bp). Wild-type allele is detected using PCR primers as follows: WT-F, 5′-CTGGTCTCATCCTCAACTCA-3′ and WT-R, 5′-TGTAACCCTAGGATTG TAGG-3′ (product size, 1563 bp).

## Reaggregate thymus experiments

Fetal thymus lobes isolated from RelB-KO mice at embryonic day (E) 15 were organ cultured for 5 days on sponge-supported Nucleopore filters (Whatman) placed on Dulbecco's modified Eagle medium supplemented with 10% fetal bovine serum and penicillin–streptomycin mixed solution. To eliminate endogenous hematopoietic cells from the lobes, 1.35 mM 2'-deoxyguanosine (dGuo) was added in the culture medium (*Anderson et al., 1993*; *White et al., 2008*). dGuo-treated thymus lobes were digested with 0.25% trypsin supplemented with 0.45 mM EDTA (ethylenediaminetetraacetic acid) and 0.02% DNase I for 15 min at 37 °C and reaggregated with tdTomato$^+$ or tdTomato$^-$ TECs isolated from *Ccl21a*$^{tdTomato/+}$ mice at E17 or 4 weeks old. After 1-day culture on Nucleopore filters, reaggregated thymuses were transplanted under kidney capsules of B6 mice. dGuo-treated fetal thymus lobes isolated from RelB-deficient and RelB-heterozygous mice were also transplanted under kidney capsules of B6 mice. Grafts were harvested 5 weeks post transplantation.

## Multicolor immunofluorescence analysis

Paraformaldehyde-fixed frozen tissues embedded in OCT compound (Sakura Finetek) were sliced into 10-μm-thick sections. The sections were stained with anti-CCL21 (Bio-Rad, Cat# AAM27, RRID:AB_2072089), anti-Aire (eBioscience, Cat# 50593482, RRID:AB_2574257), anti-Ly51 (BioLegend, Cat# 108312, RRID:AB_2099613), anti-Krt14 (BioLegend, Cat# 905304, RRID:AB_2616896), anti-DCLK1 (abcam, Cat# ab31704, RRID:AB_873537), anti-CD4 (BioLegend, Cat# 100533, RRID:AB_493372), anti-CD8 (BioLegend, Cat# 100701, RRID:AB_ 312740) antibodies, anti-Foxn1 antiserum (*Itoi et al., 2007*), and UEA-1 (Vector Laboratories, Cat# B1065, RRID:AB_2336766). Images were analyzed with a TCS SP8 confocal laser scanning microscope (Leica).

## Fluorescence distribution analysis

The plot profile intensity for the fluorescence image of each channel in the region of interest (ROI) was measured using ImageJ software (https://imagej.nih.gov/ij/index.html). The fluorescence intensity at each horizontal distance with the total intensity of the vertically averaged tdTomato, EGFP, or Ly51 in the ROI was calculated as 100. The sum of tdTomato or EGFP intensities for the medullary region defined by Krt14 expression or UEA1 reactivity and for the cortical region was calculated.

## Multicolor flow cytometric analysis and cell sorting

For the analysis of thymic epithelial cells (TECs), minced thymuses were digested with 0.5 unit/ml Liberase (Roche) in the presence of 0.02% DNase I (Roche). Single-cell suspensions were stained with antibodies specific for EpCAM (BioLegend, Cat# 118216, RRID:AB_1236471), CD45 (eBioscience, Cat# 48045180, RRID:AB_1518807), Ly51 (BioLegend, Cat# 108308, RRID:AB_313365, Cat# 108312, RRID:AB_2099613), Podoplanin (BioLegend, Cat# 127418, RRID:AB_2629804), I-Ab (BioLegend, Cat# 116406, RRID:AB_313725), and for the reactivity with UEA-1 (Vector Laboratories, Cat# B-1065, RRID:AB_2336766, Cat# FL-1061, RRID:AB_2336767, Cat# DL-1068). For the analysis of Aire and DCLK1, surface-stained cells were fixed in 5% formaldehyde neutral buffer solution (Nacarai Tesque), permeabilized in 1× permeabilization buffer (eBioscience), and stained with anti-Aire (eBioscience, Cat# 50593482, RRID:AB_2574257) or anti-DCLK1 (abcam, Cat# ab31704, RRID:AB_873537) antibody. For the isolation of TECs, CD45− cells were enriched in magnetic bead conjugated anti-CD45 antibody (Miltenyi Biotec). Multicolor flow cytometry and cell sorting were performed on FACSVerse, FACSAria II, and LSRFortessa (BD Biosciences).

## RNA-sequencing analysis

cDNAs were prepared by using SMART-Seq v4 Ultra Low Input RNA Kit, according to the manufacturer's protocol (Clontech). Sequencing libraries were generated by using a Nextera XT DNA Library Prep Kit, according to the manufacturer's protocol (Illumina). The concentration of libraries was measured by an ABI PRISM 7500 Real-time PCR system in combination with a Power SYBR Green PCR Master Mix (Thermo Fisher). Single-end sequencing of cDNA libraries with a read length of 51 was performed with HiSeq 1500 platform (Illumina). For RNA-sequencing analysis of CCL21$^+$RANK$^-$ and CCL21$^-$RANK$^+$ TECs isolated from *Ccl21a*$^{tdTomato}$ *Tnfrsf11a(RANK)*$^{Venus}$ embryonic E17 thymus, RNA was prepared using QIAGEN RNAeasy Plus Micro Kit with genomic DNA eliminator columns. RNA-sequencing samples were sequenced on NovaSeq_SP flowcell using Illumina TruSeq Stranded Total

RNA Library Prep with 2 × 100 bp paired-end sequencing. Data were analyzed by using CLC Genomics Workbench 12.0 (QIAGEN) with default parameters.

## Single-cell RNA-sequencing analysis

E12 embryonic pharyngeal cells, E14 embryonic CD45⁻EpCAM⁺ β5t-Venus⁺ and β5t-Venus⁻ TECs, E16 embryonic CD45⁻EpCAM⁺ β5t-Venus⁺ and β5t-Venus⁻ TECs isolated from β5t-Venus knock-in mice, and 4-week-old postnatal TEC-enriched thymus cells were employed for single-cell RNA-sequencing. 3′ mRNA-seq library was generated as outlined in the Chromium Connect platform using 3′ v3.1 Single Cell RNA-Seq chemistry reagents from 10× Genomics. Reverse transcription and cDNA amplification were performed according to the manufacturer's instructions. Libraries were sequenced on NextSeq 2000 and FASTQ files were generated from demultiplexes raw base call files using Cell Ranger 6.1.1 mkfastq. Alignment, filtering, barcode counting, and UMI (unique molecular identifier) counting were performed using Cell Ranger 6.1.1 count with mouse reference 3.0.0 (mm10 and Ensembl release 98). The resulting matrix was imported into Seurat (v4.3.0) in R v 4.3.0 to create a Seurat object. The object underwent normalization, filtering, scaling, and dimensionality reduction steps sequentially using Seurat build-in functions, with default parameters. To integrate all the filtered objects from each sample, the CCA algorithm was employed through the FindIntegrationAnchors and IntegrateData functions in Seurat. The Uniform Manifold Approximation and Projection plots were visualized using the DimPlot function of Seurat. The histograms were visualized using the ggplot2 package (v3.4.2).

## Statistical analysis

Statistical significance was assessed using the two-tailed unpaired Student's *t*-test with Welch's correction for unequal variances or one-way analysis of variance with Tukey's correction.

## Acknowledgements

We thank Drs. Takashi Amagai and Manami Itoi for providing anti-Foxn1 antibody, Dr. Thomas Boehm for providing Foxn1-GFP transgenic mice, Drs. Toyomasa Katagiri, Hitomi Kyuma, Eri Ootsu, Bao Tran, and Assiatu Crossman for technical assistance, and Drs. Alfred Singer, Richard Hodes, Mami Matsuda-Lennikov, and Jie Li for reading the manuscript. This work was supported by the Intramural Research Program ZIA BC 011806 of the National Institutes of Health, the National Cancer Institute, and the Center for Cancer Research (YT); by the JSPS KAKENHI 22K06900, the JSPS Bilateral Program 120219928, and the JST PRESTO Grant 22712940 (IO); and by the MRC Programme Grant MR/T029765/1 and the Wellcome Trust Collaborative Award SynThy 211944/Z/18/Z (GA).

## Additional information

### Funding

| Funder | Grant reference number | Author |
|---|---|---|
| National Institutes of Health | ZIA BC 011806 | Yousuke Takahama |
| Japan Society for the Promotion of Science | 22K06900 | Izumi Ohigashi |
| Japan Society for the Promotion of Science | 120219928 | Izumi Ohigashi |
| Japan Science and Technology Agency | 22712940 | Izumi Ohigashi |
| Medical Research Council | MR/T029765/1 | Graham Anderson |
| Wellcome Trust | 211944/Z/18/Z | Graham Anderson |

The funders had no role in study design, data collection, and interpretation, or the decision to submit the work for publication. For the purpose of Open Access, the authors have applied a CC BY public copyright license to any Author Accepted Manuscript version arising from this submission.

## Author contributions
Izumi Ohigashi, Conceptualization, Data curation, Formal analysis, Validation, Investigation, Writing - original draft, Writing - review and editing; Andrea J White, Mei-Ting Yang, Sayumi Fujimori, Yu Tanaka, Data curation, Formal analysis, Investigation; Alison Jacques, Data curation, Investigation; Hiroshi Kiyonari, Resources, Methodology; Yosuke Matsushita, Sevilay Turan, Michael C Kelly, Data curation, Methodology; Graham Anderson, Supervision, Investigation, Writing - original draft, Writing - review and editing; Yousuke Takahama, Conceptualization, Supervision, Validation, Investigation, Writing - original draft, Project administration, Writing - review and editing

## Author ORCIDs
Izumi Ohigashi http://orcid.org/0000-0003-0017-6957
Sayumi Fujimori http://orcid.org/0000-0002-1822-1296
Michael C Kelly http://orcid.org/0000-0003-0654-2778
Yousuke Takahama http://orcid.org/0000-0002-4992-9174

## Ethics
All mouse experiments were performed with consent from the Animal Experimentation Committee of the University of Tokushima (T2022-50), the Animal Care and Use Committee of the National Cancer Institute (ASP 21-431, ASP 21-432, and EIB-076-3), the Birmingham Animal Welfare and Ethical Review Board and UK Home Office Animal Licences (PP7518148 and PP2990911), and the Institutional Animal Care and Use Committee of RIKEN Kobe Branch (A2001-03).

Reviewer #1 (Public Review): https://doi.org/10.7554/eLife.92552.3.sa1
Reviewer #2 (Public Review): https://doi.org/10.7554/eLife.92552.3.sa2
Reviewer #3 (Public Review): https://doi.org/10.7554/eLife.92552.3.sa3
Author Response https://doi.org/10.7554/eLife.92552.3.sa4

---

# Additional files

## Supplementary files
• MDAR checklist

## Data availability
RNA-sequencing data have been deposited in the DDBJ BioProject database with the accession number PRJDB15439 and NCBI GEO database with the accession number GSE239913. Single-cell RNA-sequencing data have been deposited in NCBI GEO with the accession number GSE243180.

The following datasets were generated:

| Author(s) | Year | Dataset title | Dataset URL | Database and Identifier |
|---|---|---|---|---|
| Ohigashi I | 2024 | RNA-sequencing data of cortical thymic epithelial cells isolated from Ccl21a-Cre x CAG-loxP-EGFP mice | https://ddbj.nig.ac.jp/resource/bioproject/PRJDB15439 | The DNA Data Bank of Japan, PRJDB15439 |
| Ohigashi I, White A, Turan S, Anderson G, Takahama Y | 2023 | Gene expression profiles of Ccl21a+ and Rank+ E17 thymic epithelial cells | https://www.ncbi.nlm.nih.gov/geo/query/acc.cgi?acc=GSE239913 | NCBI Gene Expression Omnibus, GSE239913 |
| Takahama Y, Tanaka Y, Jacques A, Yang M, Kelly M | 2024 | Functionally diverse thymic medullary epithelial cells interplay to direct central tolerance | https://www.ncbi.nlm.nih.gov/geo/query/acc.cgi?acc=GSE243180 | NCBI Gene Expression Omnibus, GSE243180 |

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
