## [Editor Report · eLife assessment]

This **important** study provides new insights into the development and function of medullary thymus epithelial cells (mTEC). The authors provide **compelling** evidence to support their claims as to the differentiation and lineage outcomes of CCL21+ mTEC progenitors, which further our understanding of how central tolerance of T cells is enforced within the thymus.

---

## [Referee Report · Reviewer #1 (Public Review)]

The work by Ohigashi and colleagues addresses the developmental and lineage relationship of a newly characterized thymus epithelial cell (TEC) progenitor subset. The authors take advantage of an elegant and powerful set of experimental approaches to demonstrate that CCL21-expressing TECs appear early in thymus organogenesis and that these cells, which are centrally located, go on to give rise to medullary (m)TECs. What makes the findings intriguing is that these CCL21-expressing mTECs are a distinct subset, which do not express RANK or AIRE, and transcriptomic and lineage tracing approaches point to these cells as potential mTEC progenitor-like cells. Of note, using in vitro and in vivo precursor-product cell transfer experiments, the authors show that this subset has a developmental potential to give rise to AIRE+ self-antigen-displaying mTECs, revealing that CCL21-expressing mTECs can give rise to distinct mTEC subsets. This functional duality provides an attractive rationale for the necessary function of mTECs, which is to attract CCR7+ thymocytes that have just undergone positive selection in the thymus cortex to enter the medulla to undergo tolerance-induction against self-antigen-displaying mTECs. Overall, the work is well supported and offers new insights into the diverse functions of the medullary compartment, and how two distinct subsets of mTECs can achieve it.

---

## [Referee Report · Reviewer #2 (Public Review)]

The authors set out to discover a developmental pathway leading to functionally diverse mTEC subsets. They show that Ccl21 is expressed early during thymus ontogeny in the medullary area. Fate-mapping gives evidence for the Ccl21 positive history of Aire positive mTECs as well as of thymic tuft cells and postnatally of a certain percentage of cTECs. Therefore, the differentiation potential of Ccl21+ TECs is tested in reaggregate thymus experiments - using embryonic or postnatal Ccl21+ TECs. From these experiments, the authors conclude that at least embryonic mTECs in large part pass through a Ccl21 positive stage prior to differentiation towards an Aire expressing or tuft cell stage.

The authors are using Ccl21a as a marker for a bipotent progenitor that is detectable in the embryonic thymus and is still present at the adult stage mainly giving rise to mTECs. The choice of this marker gene is very interesting since Ccl21 expression can directly be linked to an important aspect in thymus biology: the expression of Ccl21 by cells in the thymic medulla allows trafficking of T cells into the medulla in order to undergo T cell selection. Making use of the Ccl21 detection, the authors can nicely show that cells actively expressing Ccl21 are localized throughout the medulla at an embryonic stage but also in adult thymus tissue. This suggests, that this progenitor is not accumulating at a specific area inside the medulla. This is a new finding. Moreover, the finding that a Ccl21+ progenitor population plays a functional role in thymocyte trafficking towards the medulla has not been described. Thus, Ccl21 expression may be used to localize a late bipotent progenitor in the thymic lobes. In addition, in Fig.8, the authors provide evidence that these progenitor cells have the potential to self-maintain as well as to differentiate in reaggregate experiments at E17 (not at 4 weeks of age). The first point is of great interest and importance since these cells in theory can be of therapeutic use.

Overall assessment:

The authors highlight a developmental pathway starting from a Ccl21-expressing TEC progenitor that contributes to a functionally diverse mTEC repertoire. This is a welcome addition to current knowledge of TEC differentiation.

---

## [Referee Report · Reviewer #3 (Public Review)]

In this manuscript, the authors define the developmental trajectory resulting in a diverse mTEC compartment. Using a variety of approaches, including a novel CCL21-fate mapping model, data is presented to argue that embryonic CCL21-expressing thymocyte attracting mTECs naturally convert to into self-antigen displaying mTEC subsets, including Aire+ mTECs and thymic tuft cells. Perhaps somewhat surprisingly, a large fraction of cTECs were also marked for having expressed CCL21, suggesting that there exists some conversion of mTEC (progenitors) into cTEC, a developmentally interesting observation that could be followed up later. Overall, the experimental setup, writing, and conclusions, are all outstanding. The one question I have, which may be more of a curiosity of this reviewer than a requirement for the manuscript, is whether thymocytes themselves are required for the conversion/maturation of attracting TECs to mTECs? For example, in CD3e-/- (or Rag-/-) mice, are mTECs arrested at the thymocyte attracting stage, or is the conversion process 'pre-programed'? In the same vein, do cTECs (or the immature cTECs) maintain CCL21 expression in the absence of mature thymocytes? These are not critical studies but are fairly straightforward (effort- and time-wise) that would aid in placing this process in the overall scope of thymus development.

---

## [Author Response]

The following is the authors’ response to the original reviews.

**eLife assessment**
This is an important study that provides new insights into the development and function of medullary thymus epithelial cells (mTEC). The authors provide compelling evidence to support their claims as to the differentiation and lineage outcomes of CCL21+ mTEC progenitors, which further our understanding of how central tolerance of T cells is enforced within the thymus.
**Public Reviews:**

**Reviewer #1 (Public Review):**
The work by Ohigashi and colleagues addresses the developmental and lineage relationship of a newly characterized thymus epithelial cell (TEC) progenitor subset. The authors take advantage of an elegant and powerful set of experimental approaches to demonstrate that CCL21-expressing TECs appear early in thymus organogenesis and that these cells, which are centrally located, go on to give rise to medullary (m)TECs. What makes the findings intriguing is that these CCL21-expressing mTECs are a distinct subset, which do not express RANK or AIRE, and transcriptomic and lineage tracing approaches point to these cells as potential mTEC progenitor-like cells. Of note, using in vitro and in vivo precursor-product cell transfer experiments, the authors show that this subset has a developmental potential to give rise to AIRE+ self-antigen-displaying mTECs, revealing that CCL21-expressing mTECs can give rise to distinct mTEC subsets. This functional duality provides an attractive rationale for the necessary function of mTECs, which is to attract CCR7+ thymocytes that have just undergone positive selection in the thymus cortex to enter the medulla to undergo tolerance-induction against self-antigen-displaying mTECs. Overall, the work is well supported and offers new insights into the diverse functions of the medullary compartment, and how two distinct subsets of mTECs can achieve it.
**Reviewer #2 (Public Review):**
Summary:The authors set out to discover a developmental pathway leading to functionally diverse mTEC subsets. They show that Ccl21 is expressed early during thymus ontogeny in the medullary area. Fate-mapping gives evidence for the Ccl21 positive history of Aire positive mTECs as well as of thymic tuft cells and postnatally of a certain percentage of cTECs. Therefore, the differentiation potential of Ccl21+ TECs is tested in reaggregate thymus experiments - using embryonic or postnatal Ccl21+ TECs. From these experiments, the authors conclude that at least embryonic mTECs in large part pass through a Ccl21 positive stage prior to differentiation towards an Aire expressing or tuft cell stage.The authors are using Ccl21a as a marker for a bipotent progenitor that is detectable in the embryonic thymus and is still present at the adult stage mainly giving rise to mTECs. The choice of this marker gene is very interesting since Ccl21 expression can directly be linked to an important aspect in thymus biology: the expression of Ccl21 by cells in the thymic medulla allows trafficking of T cells into the medulla in order to undergo T cell selection.Making use of the Ccl21 detection, the authors can nicely show that cells actively expressing Ccl21 are localized throughout the medulla at an embryonic stage but also in adult thymus tissue. This suggests, that this progenitor is not accumulating at a specific area inside the medulla. This is a new finding.Moreover, the finding that a Ccl21+ progenitor population plays a functional role in thymocyte trafficking towards the medulla has not been described. Thus, Ccl21 expression may be used to localize a late bipotent progenitor in the thymic lobes.In addition, in Fig.8, the authors provide evidence that these progenitor cells have the potential to self-maintain as well as to differentiate in reaggregate experiments at E17 (not at 4 weeks of age). The first point is of great interest and importance since these cells in theory can be of therapeutic use.Overall assessment:The authors highlight a developmental pathway starting from a Ccl21-expressing TEC progenitor that contributes to a functionally diverse mTEC repertoire. This is a welcome addition to current knowledge of TEC differentiation.
**Reviewer #3 (Public Review):**
In this manuscript, the authors define the developmental trajectory resulting in a diverse mTEC compartment. Using a variety of approaches, including a novel CCL21-fate mapping model, data is presented to argue that embryonic CCL21-expressing thymocyte attracting mTECs naturally convert to into self-antigen displaying mTEC subsets, including Aire+ mTECs and thymic tuft cells. Perhaps somewhat surprisingly, a large fraction of cTECs were also marked for having expressed CCL21, suggesting that there exists some conversion of mTEC (progenitors) into cTEC, a developmentally interesting observation that could be followed up later. Overall, the experimental setup, writing, and conclusions, are all outstanding.

Provisional author response

We thank the editors and the reviewers for their supportive comments on our manuscript. We will revise the manuscript according to their helpful recommendations.

Author response to recommendations

We thank the editors and the reviewers for their supportive comments on our manuscript. We also thank the three reviewers for their helpful recommendations. We have revised the manuscript accordingly, as detailed below.

**Reviewer #1 (Recommendations For The Authors):**
There are several unanswered questions, which the authors themselves acknowledge, a principal one being whether CCL21+ mTECs represent a progenitor for yet another distinct subset of cortical (c)TECs, or whether they represent an intermediary or unique population of mTECs derived from a bipotent (cTEC/mTEC) progenitor. These questions will need to be addressed in future work as they go beyond the initial characterization of this intriguing mTEC subset.

Indeed, our findings reported in this manuscript have stimulated many interesting questions, including those pointed out by the reviewer. We would like to address them one by one in our future work.

The presence of GFP+ cTECs, which are lineage-traced as having expressed CCL21, begs the question as to whether these cells are generated as a consequence of later steps in mTEC differentiation or derived from earlier bipotent cells, which again the authors point out. The authors could discuss this further or perhaps experimentally address this by using a model system whereby mTEC differentiation is absent or halted (e.g., Relb ko, or TCRa/TCRd ko) and test whether GFP+ cTECs are still present.

According to the suggestion, we have revised the manuscript by adding a statement that it is interesting to examine whether GFP+ cTEC development in Ccl21a-Cre x CAG-loxP-EGFP mice is mediated through RelB-dependent mTEC developmental progression or developing thymocyte-dependent mTEC-nurturing ‘crosstalk’ signals.

**Reviewer #2 (Recommendations For The Authors):**
Even though the manuscript highlights the functional aspect of a postnatal bipotent progenitor, there are several aspects that need further discussion.(1) The title is somewhat misleading since the identified TEC subset can not only be detected in embryonic, but also in postnatal thymus. Only the RTOC experiments indicate a higher developmental potential of TECs isolated from embryos, but this might as well be due to experimental difficulties as discussed in the text. Furthermore, Ccl21+ TECs are shown to differentiate postnatally into mTECs and cTECs, therefore this subset presumably belongs to a bipotent progenitor population described earlier (their ref. 22, 39).

We are fully aware of previous studies showing that mTEC progenitors include cells that transcribe Ccl21a, and have cited them in the manuscript. The manuscript title describes our finding that thymocyte-attracting CCL21-expressing functional mTECs isolated from embryonic thymus show the capability to give rise to self-antigen-displaying mTECs. We thank the reviewer for further pointing out the possibility that postnatal CCLl21+ TECs include cells that retain the capability to differentiate into mTECs and cTECs.

(2) In the introduction the authors claim that the "developmental progression of the self-antigen-displaying mTEC subset occurs in a single stream as mTEClow progenitors -> mTEChigh Aire-expressing cells -> mTEClow mimetic cells." line 79. So far it only could be shown that some mimetic cell types undergo an Aire+ stage; whether this is true for all mimetic cells remains to be shown. Therefore, this statement should be toned down.

Following the suggestion, this sentence has been toned down in the revised manuscript.

(3) In line 86, the reference to another paper, describing Ccl21a expression in a postnatal mTEC biased progenitor should be added: Nusser et al. Nature. 2022 PMID: 35614226, in which the developmental potential of the Ccl21 positive so-called postnatal progenitor is analysed by barcoding and results give evidence for differentiation into mature mTECs (see lines 94-96).

As suggested, the Introduction of the revised manuscript now cites Nusser, et al. study showing that postnatal mTEC-biased progenitors include cells that transcribe Ccl21a.

(4) Have a look at Extended Data Figure 2b of PMID: 35614226, wherein the population-specific gene expression pattern of the progenitor population at different time points is depicted. Ccl21a belongs to a group of genes, which identifies the postnatal progenitor, and indicates that its functionality and/or developmental potential is age-dependent. Therefore, it would be important to specify the age of the analysed mice throughout the text of the results part instead of describing them as "postnatal" only.

As recommended, mouse age has been added to the revised manuscript and figures.

(5) Line 113: "embryonic" needs to be replaced since the results of Fig. 1 are referring to 5-week-old mice.

The manuscript has been revised per the reviewer’s suggestion.

(6) Referring to Fig. 3g, line 173: It is interesting to see that, at 3 weeks of age, 95% of mTECs have a Ccl21-history but only approx. 70% of cTECs. Therefore, the earliest progenitor giving rise to the first cTECs might still be productive and feed into the cTEC lineage. This reporter would allow for the analysis of progenitor activity over time. The same could be done for mTECs since at E15 the tdTomato signal is still low compared to the assigned medullary area in Fig. 2c in order to detect when the Ccl21-expressing progenitor becomes the main source of mTECs. The finding in Fig. 4e (line196) also argues for the timed replacement of cTECs by a progenitor which locates to the medulla, thus, leading to a decline in Ccl21-history signal towards the subcapsular region at 2 weeks of age. This should be better explained/discussed.

We appreciate the work of Nusser, et al. showing that postnatal mTEC-biased, but not embryonic cTEC-biased, TEC progenitors include cells that transcribe a detectable amount of Ccl21a (cited in the Introduction as ref. 23). It is important to clarify whether and how those postnatal TEC progenitors (23) overlap with the embryonic and postnatal CCL21-protein-expressing mTECs reported in this study. It is also interesting to shed light on how Ccl21a+ progenitors contribute to cTECs and mTECs over the ontogeny and whether the enrichment of Ccl21a+ progenitor-derived cTECs in the perimedullary area reflects a temporal replacement of cTECs derived from Ccl21a+ progenitors localized in the medulla. We would like to clarify these issues in our future work. The revised manuscript includes a discussion of these issues.

(7) Line 304 and 355: Note that the "unstable" age-dependent gene expression profiles were already reported in Nusser et al. Nature. 2022. Not only Ccl21 expression, but other progenitor-specific genes also change their expression levels with age. The entirety of changes in gene expression during aging likely impacts the developmental potential of progenitor populations. These changes might be reflected in the negative results of the RTOC experiment using TECs of 4-week-old mice. The manuscript would benefit from a discussion in light of this "unstable" age-dependent gene expression.

It is interesting to point out that the age-dependent difference in gene expression profiles, which was reported in TEC progenitors by Nusser, et al. (23), is also detected in CCL21-expressing mTECs in this study. Similarly to the recommendation no. 6 by reviewer 2, and as described in the revised manuscript, it is interesting to clarify whether and how embryonic and postnatal CCL21-expressing mTECs overlap with the previously reported TEC progenitors.

(8) Line 321: as discussed above, the exact time point should be added to the text since the proportion of cTECs derived from a Ccl21+ progenitor is associated with a certain time point, "2/3 of cTECs" refers to 3 weeks of age.

The manuscript has been revised following the reviewer’s suggestion.

**Reviewer #3 (Recommendations For The Authors):**
The one question I have, which may be more of a curiosity of this reviewer than a requirement for the manuscript, is whether thymocytes themselves are required for the conversion/maturation of attracting TECs to mTECs? For example, in CD3e-/- (or Rag-/-) mice, are mTECs arrested at the thymocyte attracting stage, or is the conversion process 'pre-programed'? In the same vein, do cTECs (or the immature cTECs) maintain CCL21 expression in the absence of mature thymocytes? These are not critical studies but are fairly straightforward (effort- and time-wise) that would aid in placing this process in the overall scope of thymus development.

We previously showed that Aire+ mTECs are detectable in the thymus of RAG2-deficient mice, in which thymocyte development is arrested beyond the CD4/CD8 double-negative 3 stage (Hikosaka, et al. 2006; PMID: 18799150). In another work, we also showed that Aire+ mTECs and CCL21+ mTECs are detectable in the thymus of TCR-alpha-KO mice, which lack mature CD4/CD8 single-positive TCR-alpha/beta-expressing thymocytes (Lkhagvasuren, et al. 2013; PMID: 23585674). These results indicate that thymocyte maturation beyond the Rag-dependent stage is not essential for the development of Aire+ mTECs. Nonetheless, we agree with the reviewer pointing out that it is important to clarify how developing thymocytes contribute to the growth and differentiation of diverse TEC subpopulations, including GFP+ cTEC development in Ccl21a-Cre x CAG-loxP-EGFP mice. The revised manuscript includes a discussion of these issues.